# GABA$_B$ receptor auxiliary subunits modulate Cav2.3-mediated release from medial habenula terminals

Pradeep Bhandari[1], David Vandael[1], Diego Fernández-Fernández[2], Thorsten Fritzius[2], David Kleindienst[1], Cihan Önal[1], Jacqueline Montanaro[1], Martin Gassmann[2], Peter Jonas[1], Akos Kulik[3,4], Bernhard Bettler[2], Ryuichi Shigemoto[1]*, Peter Koppensteiner[1]*

[1]Institute of Science and Technology Austria (IST Austria), Klosterneuburg, Austria; [2]Department of Biomedicine, University of Basel, Basel, Switzerland; [3]Institute of Physiology II, Faculty of Medicine, Freiburg, Germany; [4]BIOSS Centre for Biological Signalling Studies, University of Freiburg, Freiburg, Germany

**Abstract** The synaptic connection from medial habenula (MHb) to interpeduncular nucleus (IPN) is critical for emotion-related behaviors and uniquely expresses R-type Ca$^{2+}$ channels (Cav2.3) and auxiliary GABA$_B$ receptor (GBR) subunits, the K$^+$-channel tetramerization domain-containing proteins (KCTDs). Activation of GBRs facilitates or inhibits transmitter release from MHb terminals depending on the IPN subnucleus, but the role of KCTDs is unknown. We therefore examined the localization and function of Cav2.3, GBRs, and KCTDs in this pathway in mice. We show in heterologous cells that KCTD8 and KCTD12b directly bind to Cav2.3 and that KCTD8 potentiates Cav2.3 currents in the absence of GBRs. In the rostral IPN, KCTD8, KCTD12b, and Cav2.3 co-localize at the presynaptic active zone. Genetic deletion indicated a bidirectional modulation of Cav2.3-mediated release by these KCTDs with a compensatory increase of KCTD8 in the active zone in KCTD12b-deficient mice. The interaction of Cav2.3 with KCTDs therefore scales synaptic strength independent of GBR activation.

*For correspondence:
ryuichi.shigemoto@ist.ac.at (RS);
peter.koppensteiner@ist.ac.at (PK)

Competing interests: The authors declare that no competing interests exist.

## Introduction

The medial habenula (MHb) is an epithalamic structure that exclusively projects to the interpeduncular nucleus (IPN), with the dorsal MHb projecting to the lateral IPN and the ventral MHb projecting to the rostral and central subnuclei of the IPN (*Figure 1A*). This pathway is involved in various behaviors, including nicotine addiction and aversion (*Agetsuma et al., 2010*; *Koppensteiner et al., 2016*; *Koppensteiner et al., 2017*; *Melani et al., 2019*; *Zhang et al., 2016*; *Zhao-Shea et al., 2013*). A striking property of the MHb-IPN pathway is the prominent presynaptic localization of the R-type voltage-gated Ca$^{2+}$ channel 2.3 (Cav2.3), a channel mainly located in postsynaptic elements in other brain areas (*Parajuli et al., 2012*). Furthermore, activation of presynaptic GABA$_B$ receptors (GBRs) on MHb terminals exerts an unusual facilitatory effect by increasing neurotransmitter release up to 10-fold (*Zhang et al., 2016*), and this effect appears to be involved in synaptic plasticity (*Koppensteiner et al., 2017*). Interestingly, this potentiation via presynaptic GBR activation only occurs in the glutamatergic/cholinergic pathway from the ventral MHb to the rostral/central IPN, whereas in the glutamatergic/substance-P-ergic pathway from the dorsal MHb to the lateral IPN, the release is attenuated by GBR activation (*Melani et al., 2019*). Although GBRs are ubiquitously expressed in these MHb-IPN pathways (*Margeta-Mitrovic et al., 1999*), the presynaptic localization and function of their auxiliary KCTD subunits is unknown.

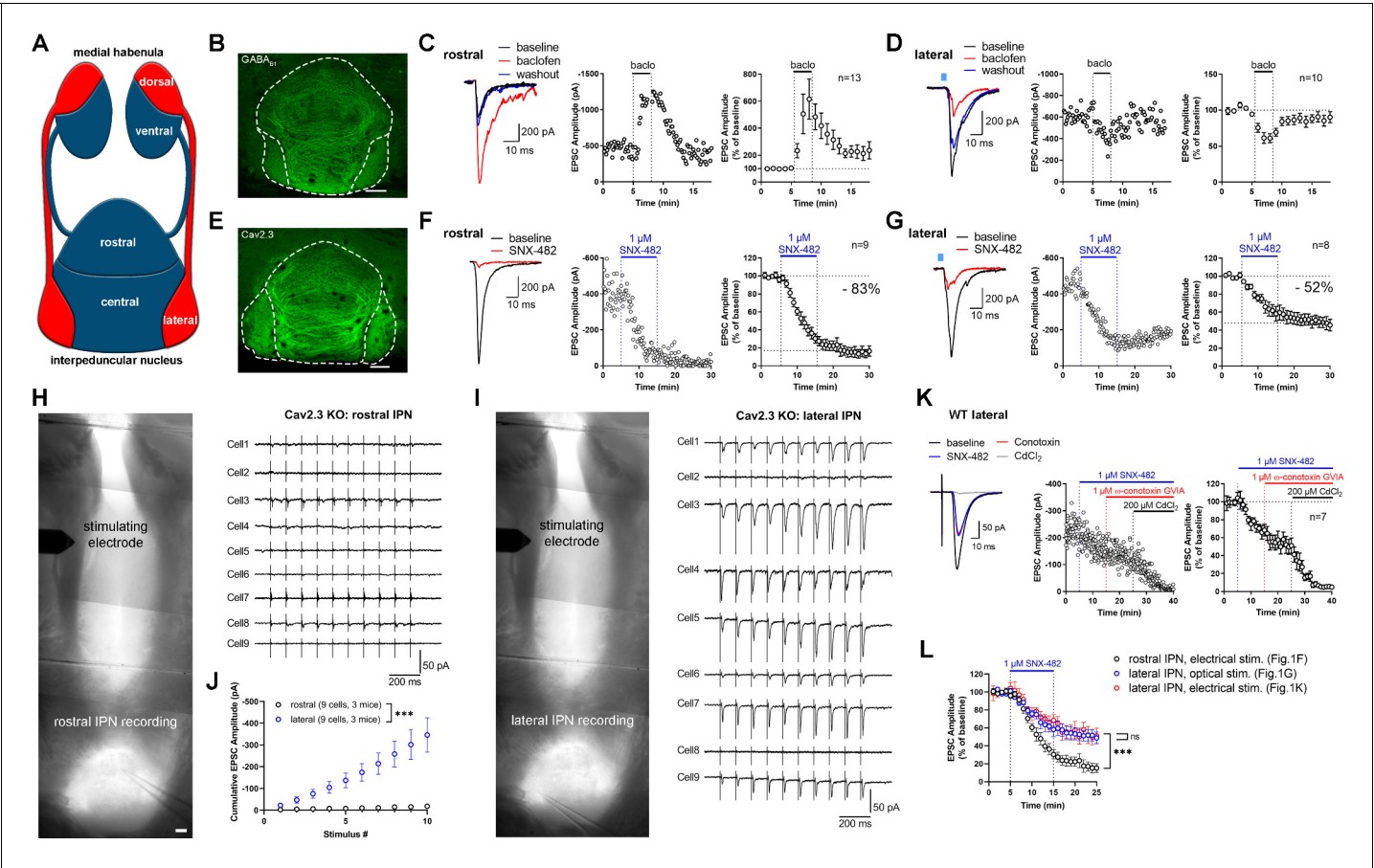

**Figure 1.** Expression and function of GABA_B receptors and Cav2.3 at two parallel MHb-IPN pathways. (A) Schematic drawing of the two MHb-IPN pathways. In red: the dorsal part of the MHb projects to the lateral subnuclei of the IPN. In blue: the ventral part of the MHb projects to the rostral/central subnuclei of the IPN. (B) Confocal image of GABA_B1 immunofluorescence signal indicates the presence of GABA_B receptors (GBRs) in all IPN subnuclei. (C) In whole-cell recordings of rostral IPN neurons, activation of GBRs by baclofen (1 μM) produced a potentiation of electrically evoked EPSC amplitudes. Left: example EPSC traces before (black) and during the application of baclofen (red) and after washout of baclofen (blue); middle: example time course of EPSC amplitudes in one cell; right: averaged time course of relative EPSC amplitude change after baclofen (n = 13 cells/9 mice). (D) Baclofen reduced the amplitude of light-evoked glutamatergic EPSCs in lateral IPN neurons (n = 10 cells/5 mice). (E) Confocal image of Cav2.3 immunofluorescence signal indicates Cav2.3 presence in MHb axonal projections of both MHb-IPN pathways. (F) Pharmacological inhibition of Cav2.3 with SNX-482 in whole-cell recordings of rostral IPN neurons. Left: example traces before and after the application of SNX-482; middle: example time course of EPSC amplitude reduction by SNX-482; right: averaged time course of relative EPSC amplitude reduction by SNX-482. EPSC amplitudes were reduced by 83% on average (n = 9 cells/9 mice). (G) In Tac1-ChR2-EYFP mice, SNX-482 reduced light-evoked glutamatergic EPSC amplitudes on average by 52% (n = 8 cells/4 mice). (H, I) Left: Positions of recording and stimulating electrodes in acute thick-slice preparations from Cav2.3 KO mice. The stimulating electrode was placed on the fasciculus retroflexus just below the MHb, 2–3 mm from the recording sites. The position of the stimulating electrode and the stimulation intensity remained unchanged between rostral and lateral IPN recordings. Right: 10 Hz EPSC traces of all recorded neurons (9 cells/3 mice). (J) Cumulative EPSC amplitude plot shows significantly higher EPSC amplitudes in lateral compared to rostral IPN neurons. ***p<0.0001 two-way ANOVA; (K) FR stimulation and lateral IPN recording in 1 mm thick slice of wild-type mice. Sequential application of SNX-482, ω-conotoxin GVIA, and CdCl2. (L) Time course overlay of (F), (G), and (K) to compare SNX-482 time courses between minutes 0 and 25. For simplicity, application of ω-conotoxin GVIA starting from minute 15 in (K) not indicated in the graph. ***p<0.0001 two-way ANOVA with Tukey's post hoc test. Scale bars in(B), (E), (H), and (I) are 100 μm. Averaged data is presented as mean ± SEM. See also *Figure 1—figure supplement 1*.

The online version of this article includes the following source data and figure supplement(s) for figure 1:

**Source data 1.** Expression and function of GABA_B receptors and Cav2.3 at two parallel MHb-IPN pathways.

**Figure supplement 1.** Effect of SNX-482 on action potential properties and confirmation of pathway specificity in thick slice recordings.

**Figure supplement 1—source data 1.** Effect of SNX-482 on action potential properties and confirmation of pathway specificity in thick slice recordings.

Four KCTD subunits, KCTD8, 12, 12b, and 16, serve as GBR auxiliary subunits in the brain (*Schwenk et al., 2010*). KCTD subunits bind as hetero- and homo-pentamers to GBRs and modulate their signaling kinetics (*Fritzius et al., 2017*; *Zheng et al., 2019*). Structurally, KCTD subunits are composed of an N-terminal T1 domain and an H1 domain. The T1 domain binds to the GABA$_{B2}$ subunit, while the H1 domain binds to the Gβ subunit of the guanine nucleotide-binding protein (G-protein) (*Schwenk et al., 2010*; *Turecek et al., 2014*; *Zheng et al., 2019*). X-ray crystallography of the T1 and H1 domains of KCTD12 and KCTD16 (*Lepore et al., 2019*; *Zheng et al., 2019*; *Zuo et al., 2019*) revealed that the secondary and tertiary protein structures of those domains are highly conserved in the KCTD proteins. However, the T1 domain possesses a highly divergent loop between helix α2 and strand β3 (following the nomenclature from *Zuo et al., 2019*), while the H1 domain features a divergent loop between the strands β4 and β5 (following the nomenclature from *Zheng et al., 2019*). These loops are hydrophilic, exposed at the protein surface, and potentially allow the different KCTD subunits to bind proteins other than GABA$_{B2}$ or Gβ. KCTD8 and 16, but not KCTD12 and 12b, additionally encode a C-terminal H2 domain. The absence of the H2 domain enables KCTD12 and 12b to rapidly desensitize GBR responses by uncoupling the Gβγ subunits of the G-protein from effector channels (*Turecek et al., 2014*). KCTD12 and 12b can form hetero-pentamers with each other but also with KCTD8 and 16 (*Fritzius et al., 2017*). Compared to the rest of the brain, the expression patterns of KCTD subunits in the MHb are unique: With the exception of a weak expression in the cerebellum and superior colliculus, KCTD8 is exclusively and strongly expressed in MHb and, to a lesser extent, IPN neurons. Furthermore, KCTD12b is exclusively expressed in the ventral part of the MHb (*Metz et al., 2011*). In contrast, KCTD12 is weakly expressed in the ventral part of the MHb, while KCTD16 is expressed in most brain areas but not the MHb. Based on proteomics studies, voltage-gated Ca$^{2+}$ channels co-precipitate GBRs and their auxiliary KCTD subunits together with release machinery proteins of the presynaptic active zone, including SNAP-25, synaptotagmins, synaptobrevin-2, Munc13-1, syntaxins, RIM1, and synapsins, among others (*Müller et al., 2010*). Furthermore, KCTD8 and KCTD16 were consistently found to co-purify with presynaptic Cav2.2 Ca$^{2+}$ channels (*Müller et al., 2010*; *Schwenk et al., 2016*). However, the functional consequences of these interactions and whether other voltage-gated Ca$^{2+}$ channels interact with KCTDs remain unknown.

Here, we studied the nano-anatomy of Cav2.3, GBRs, and KCTDs and their roles in the modulation of neurotransmission from the MHb to the IPN. Our results demonstrate that Cav2.3 is located in the presynaptic active zone of habenular terminals and required for fast neurotransmitter release. In heterologous cells, we found that KCTD8 and KCTD12b, but not KCTD12, bind to Cav2.3 at the plasma membrane. Furthermore, Cav2.3 currents were enhanced by co-expression of KCTD8, but not KCTD12b in the absence of GBRs. Strikingly, genetic deletion of KCTD8 or KCTD12b reduced or increased, respectively, the probability of neurotransmitter release in the ventral MHb to the rostral IPN pathway without affecting GBR-mediated potentiation. The increase of release in KCTD12b knock-out (KO) mice was accompanied by a compensatory recruitment of KCTD8 into the active zone, where it closely associates with Cav2.3. Furthermore, viral expression of KCTD12b in MHb neurons reduced the increased release probability in KCTD12b KO mice, whereas overexpression of KCTD8 in MHb neurons of wild-type mice increased release probability. These results support that synaptic strength at the MHb-IPN pathway is scaled via GBR-independent Cav2.3–KCTD interactions in the presynaptic active zone.

## Results

### Cav2.3 mediates neurotransmission in two distinct MHb–IPN pathways

The MHb to IPN pathway comprises two major projections (*Figure 1A*). The dorsal MHb projects to the lateral subnuclei of the IPN and releases glutamate and substance P, whereas the ventral MHb projects to the rostral and central IPN subnuclei and co-releases glutamate and acetylcholine (*Aizawa et al., 2012*; *Melani et al., 2019*; *Molas et al., 2017*; *Ren et al., 2011*). In confocal light microscopy, immunofluorescence signal for GBRs was detected in all IPN subnuclei (*Figure 1B*). GBR activation has been reported to facilitate both electrically and optogenetically evoked neurotransmitter release in the ventral MHb-rostral/central IPN pathway (*Koppensteiner et al., 2017*; *Zhang et al., 2016*), whereas it inhibits release in the dorsal MHb-lateral IPN (*Melani et al., 2019*).

We confirmed that rostral IPN neurons exhibit a strong increase in excitatory postsynaptic current (EPSC) amplitudes following GBR activation with 1 µM baclofen (*Figure 1C*), whereas EPSC amplitudes were reduced by baclofen in lateral IPN neurons (*Figure 1D*). Both the rostral/central and lateral subnuclei showed prominent Cav2.3 immunofluorescence signals (*Figure 1E*), in accordance with a previous study showing strong and exclusive presynaptic Cav2.3 localization in the IPN (*Parajuli et al., 2012*). A previous report tested the functional involvement of Cav2.3 in neurotransmission from the ventral MHb to the IPN by applying $Ni^{2+}$ (*Zhang et al., 2016*), an ion known to also inhibit other $Ca^{2+}$ channels (*Lee et al., 1999*). To test Cav2.3 dependency of neurotransmission in both pathways, we performed whole-cell recordings from IPN neurons in acute brain slices and applied the Cav2.3 blocker SNX-482 (*Newcomb et al., 1998*). In rostral and lateral IPN neurons, 1 µM SNX-482 strongly reduced the amplitude of EPSCs by 83% and 52%, respectively (*Figure 1F,G*). The SNX-482-mediated reduction in EPSC amplitudes was significantly stronger in the ventral MHb to rostral IPN pathway compared to the dorsal MHb to lateral IPN pathway (main effect of IPN subnucleus $F_{1, 395} = 213.7$, p<0.0001; two-way ANOVA). In addition, SNX-482 did not significantly affect action potential peak times or full-widths at half maximum in ventral MHb neurons (*Figure 1—figure supplement 1A*; peak times: control: $0.86 \pm 0.03$ ms, n = 3; SNX-482: $0.93 \pm 0.05$ ms, n = 4; $t_5 = 1.083$, p=0.3282; full-widths: control $1.08 \pm 0.07$ ms, n = 3; SNX-482: $1.06 \pm 0.03$ ms, n = 4, $t_5 = 0.3852$, p=0.7159; unpaired t-test), suggesting that the reduction of neurotransmitter release by SNX-482 was not caused by a blockade of voltage-gated $Na^+$ or $K^+$ channels. To corroborate the involvement of Cav2.3 in release from MHb terminals, we next performed electrical stimulation of the MHb-derived fiber tract, the fasciculus retroflexus (FR), in Cav2.3 KO mice and recorded EPSCs in rostral and lateral IPN neurons. To obtain maximally intact FR, slices were cut at 1 mm thickness, which allowed for the stimulation of the fiber tract at a distance of 2–3 mm from the IPN (*Figure 1H, I*). The MHb is a ventricular structure and therefore well supplied with freshly oxygenated ACSF, even in thick slices, assuring long-lasting viability of intact axons and their terminals within the IPN. In wild-type (WT) animals, electrical stimulation of the FR in slices prepared this way produced reliable responses in all recorded rostral and lateral IPN neurons (*Figure 1—figure supplement 1B,D*). Pathway specificity was confirmed by baclofen, which enhanced responses in the rostral but depressed responses in the lateral IPN (*Figure 1—figure supplement 1C–D*). In order to compare responses between rostral and lateral IPN recordings in Cav2.3 KO mice, the position of the stimulating electrode and the stimulus intensity remained unchanged between recordings. From each KO animal, three neurons each from the rostral and lateral IPN were recorded in an alternating manner (*Figure 1H,I*). Responses were approved when (1) evoked events were observed in response to at least six consecutive stimuli in the train, (2) the amplitude of evoked responses exceeded three times the standard deviation of the baseline noise, and (3) responses showed the characteristic shapes of a fast rising and slowly decaying synaptic response. Applying these criteria, 10 Hz electrical stimulation for 1 s failed to evoke synaptic responses in the nine of nine recorded rostral IPN neurons (*Figure 1H,J*). In contrast, seven of nine recorded lateral IPN neurons showed EPSC responses (*Figure 1I,J*; main effect of IPN subnucleus: $F_{1, 160} = 120.4$, p<0.0001; two-way ANOVA).

To investigate the identity of additional $Ca^{2+}$ channels involved in release from dorsal MHb terminals, we performed FR stimulation to measure EPSCs in lateral IPN neurons (seven recordings/five mice) in WT mice and sequentially applied SNX-482, the Cav2.2 blocker ω-conotoxin GVIA (1 µM) and $CdCl_2$ (200 µM) (*Figure 1K*). While SNX-482 produced a similar decrease in EPSC amplitude as in the optogenetic stimulation experiment (*Figure 1K,L*, effect of electrical vs. optogenetic stimulation: $F_{1, 305} = 1.830$, p=0.1772; main effect of SNX: $F_{23, 305} = 17.41$, p<0.0001; Interaction $F_{23, 305} = 1.842$, p>0.9999, two-way ANOVA of time courses between minutes 0–25), ω-conotoxin did not reduce EPSC amplitudes further. The remaining EPSC amplitude was completely blocked by $CdCl_2$, suggesting an involvement of either P/Q- or L-type $Ca^{2+}$ channels. Comparison of the SNX-482 effects on electrically and optogenetically evoked EPSCs in the lateral IPN with that on electrically evoked EPSCs in the rostral IPN confirmed a significantly weaker effect in dorsal MHb terminals in the lateral IPN than in ventral MHb terminals in the rostral IPN (*Figure 1L*; electrical stim. rostral vs. electrical stim. lateral: p<0.0001; electrical stim. rostral vs. optogenetic stim. lateral: p<0.0001; electrical stim. lateral vs. optogenetic stim. lateral p=0.3174; two-way ANOVA with Tukey's post hoc test). Overall, these results suggest that ventral MHb terminals in the rostral IPN rely exclusively on Cav2.3 for release, whereas dorsal MHb terminals in the lateral IPN rely on Cav2.3 and additional $Ca^{2+}$ channels.

# KCTD8 and KCTD12b directly bind Cav2.3 and KCTD8 enhances currents through Cav2.3 in HEK cells

GBRs strongly modulate release in both pathways. Therefore, we next investigated the expression of GBR auxiliary subunits, KCTDs, in the IPN. Immunoreactivity for KCTD8 appeared strong in all IPN subnuclei (*Figure 2A*), consistent with its strong expression throughout MHb and presynaptic localization in IPN. KCTD12 immunofluorescence appeared strong in the rostral and central subnuclei but faint in the lateral subnucleus (*Figure 2B*). In contrast, immunoreactivity for KCTD12b was only present in the rostral and central but not the lateral IPN subnuclei, indicating its absence in the dorsal MHb-IPN pathway (*Figure 2C*). While KCTD12 immunofluorescence patterns in the rostral region suggested a mostly postsynaptic expression, KCTD12b signal showed the characteristic pattern of cholinergic MHb axons inside the rostral/central IPN, suggesting presynaptic expression. Antibody

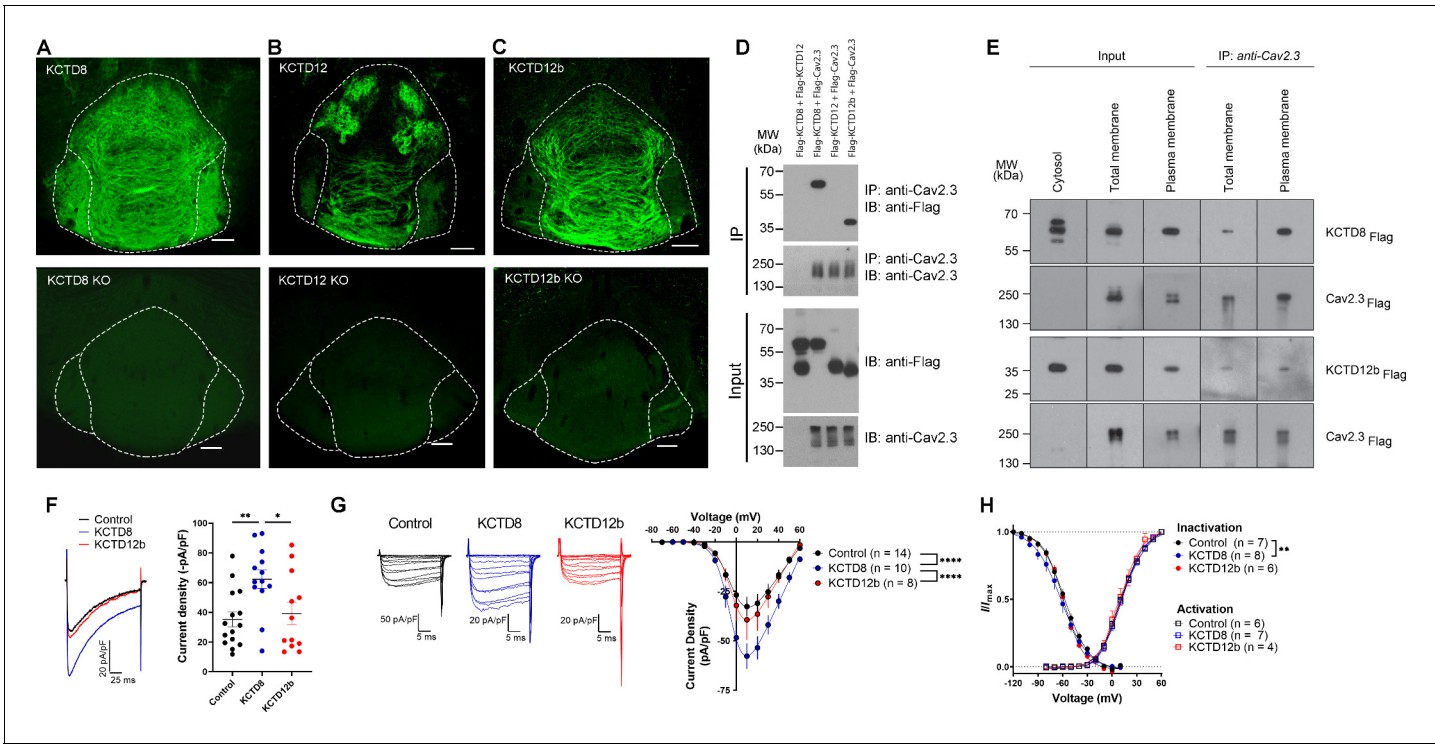

**Figure 2.** KCTD subtype expression in the IPN and interaction of Cav2.3 with KCTD8 and KCTD12b in vitro. (**A–C**) Confocal images of immunofluorescence signals of KCTD8, KCTD12, and KCTD12b in the IPN of WT (upper panels) and respective KO mice (lower panels). KCTD8 immunofluorescence was present in all IPN subnuclei, whereas KCTD12 and KCTD12b signals were observed only in the rostral/central but not the lateral IPN subnuclei. Scale bars: 100 μm. (**D**) Co-immunoprecipitation from total cell lysates of HEK293T cells transfected with Flag-tagged KCTDs and Cav2.3. Immunoprecipitation of Cav2.3 co-precipitated KCTD8 and KCTD12b, but not KCTD12. Input lanes (bottom) indicate expression of the tagged proteins in the cell lysates. (**E**) KCTDs are co-localized and interact with Cav2.3 at the cell surface of HEK293T cells. The three input lanes to the right show expression of Flag-tagged KCTD8 (top) and Flag-tagged KCTD12b (bottom) in the cytosol, the total membrane fraction ('total membrane') and the plasma membrane fraction, from left to right. The two IP lanes to the left show that immunoprecipitation of Cav2.3 in the total membrane fraction ('total membranes') and the plasma membrane fraction co-precipitated KCTD8 (top) and KCTD12b (bottom), from left to right. Membrane-bound Cav2.3 (bottom lanes) is expressed in the total membrane fraction and the plasma membrane fraction, but absent from the cytosol fraction. (**F**) Whole-cell recordings from HEK293 cells stably expressing Cav2.3. $Ba^{2+}$ current densities measured in response to a single depolarizing voltage step from −80 to 10 mV were significantly increased in KCTD8 co-transfected cells. *$p<0.05$, **$p<0.01$ one-way ANOVA with Tukey post hoc test. (**G**) Current density-to-voltage relationship demonstrating higher current densities in KCTD8-transfected cells compared with Control- and KCTD12b-transfected cells. ****$p<0.0001$ two-way ANOVA with Tukey post hoc test. (**H**) Activation and inactivation curves in Control-, KCTD8-, and KCTD12b-transfected cells. **$p<0.01$, two-way ANOVA with Tukey post hoc test. See also *Figure 2—figure supplement 1*.

The online version of this article includes the following source data and figure supplement(s) for figure 2:

**Source data 1.** KCTD subtype expression in the IPN and interaction of Cav2.3 with KCTD8 and KCTD12b in vitro.

**Figure supplement 1.** Quantification of KCTDs at the plasma and total membrane in HEK cells co-expressing Cav2.3, auxiliary β3 and α2δ1 subunits.

**Figure supplement 1—source data 1.** Quantification of KCTDs at the plasma and total membrane in HEK cells co-expressing Cav2.3, auxiliary β3 and α2δ1 subunits.

specificity for all KCTD antibodies was confirmed using the corresponding KO animals (*Figure 2A–C*), while the specificity of the anti-Cav2.3 and anti-GABA$_{B1}$ antibodies has been confirmed previously (*Kulik et al., 2002*; *Parajuli et al., 2012*).

A previous study positioned presynaptic Ca$^{2+}$ channels at the center of large, macromolecular complexes that also contained GBRs and KCTDs (*Müller et al., 2010*), and some KCTDs were found to co-purify with Cav2.2 subunit of N-type Ca$^{2+}$ channels in the absence of GBRs (*Schwenk et al., 2016*). To test whether presynaptic KCTDs in MHb terminals may directly interact with Cav2.3, we performed co-immunoprecipitation (co-IP) experiments in HEK293T cells transiently expressing Cav2.3 and either KCTD8, KCTD12, or KCTD12b. Interestingly, we found selective binding of Cav2.3 to KCTD8 and KCTD12b, but not to KCTD12 (*Figure 2D*). Plasma membrane preparations (see Materials and methods) revealed that in HEK293T cells co-expressing Cav2.3 and KCTDs, both KCTD8 and KCTD12b are highly and similarly enriched in plasma membrane extracts, when compared to total membrane extracts (*Figure 2—figure supplement 1*; KCTD8: 10.7-fold [±1.9], KCTD12b: 12.7-fold [±2.6], no significant difference between KCTD8 and KCTD12b, p=0.34, unpaired t-test). We next performed co-IP experiments using total and plasma membrane extracts of transfected HEK293T cells and found that both KCTD8 and KCTD12b associate with Cav2.3 at the plasma membrane (*Figure 2E*). These results suggest that KCTD8 and KCTD12b in MHb-derived axon terminals may directly interact with presynaptic Cav2.3, even in the absence of GBRs.

Using a cell line stably expressing human Cav2.3 (*Dai et al., 2008*), we next co-expressed either an empty vector (Control), KCTD8, or KCTD12b, and measured Ba$^{2+}$ currents through Cav2.3. Current densities resulting from a single voltage step (–80 mV to 10 mV) were significantly increased in cells co-expressing KCTD8 compared with control- and KCTD12b-transfected cells (*Figure 2F*, $F_{2, 37}$ = 5.614; p=0.0074; Control vs. KCTD8: p=0.0086; Control vs. KCTD12b: p=0.8959; KCTD8 vs. KCTD12b: p=0.0382; one-way ANOVA with Tukey post hoc test). The current density-to-voltage relationship showed a significant difference between KCTD8 and both Control- and KCTD12b-transfected cells (*Figure 2G*; main effect of transfection: $F_{2, 406}$ = 29.23, p<0.0001; Control vs. KCTD8: p<0.0001; Control vs. KCTD12b: p=0.5793; KCTD8 vs. KCTD12b: p<0.0001; two-way ANOVA with Tukey post hoc test; see also *Supplementary file 1*). Furthermore, there was a slight hyperpolarization of the steady-state inactivation in the KCTD8-expressing cells compared with Control (*Figure 2H*; main effect of transfection: $F_{2, 252}$ = 4.831, p=0.0087; Control vs. KCTD8: p=0.0059; Control vs. KCTD12b: p=0.2667; two-way ANOVA with Tukey post hoc test), whereas the voltage dependence of channel activation was not different between groups (*Figure 2H*; $F_{2, 201}$ = 2.303; p=0.1025, two-way ANOVA). These results suggest that the interaction of Cav2.3 with KCTD8, but not KCTD12b, increases the current through Cav2.3 in heterologous cells.

## Pre-embedding and freeze-fracture replica immunolabeling of presynaptic Cav2.3, GABA$_{B1}$, and KCTDs in habenular terminals in rostral and lateral IPN subnuclei

To obtain insight into the nanoscale distribution of presynaptic molecules, we investigated the electron microscopic localization of Cav2.3, GABA$_{B1}$, and KCTDs in ventral and dorsal MHb terminals. We performed pre-embedding immunogold labeling and quantified the number of silver-intensified particles in the active zone, peri-synaptic region (0–50 nm distance from the edge of the active zone), and extrasynaptic regions (50–100, 100–150, and 150–200 nm distance from the edge of the active zone; *Figure 3*, *Figure 3—figure supplement 1*). Immunogold particles for Cav2.3 were concentrated in the active zone and, to a lesser extent, the immediate peri-synaptic region (*Figure 3A*). In accordance with our immunohistochemical data (*Figure 2B,C*), KCTD12 and KCTD12b labeling was detected only in MHb terminals in the rostral but not lateral IPN subnuclei (*Figure 3D,E*). In addition, particles for KCTD12 were found mostly on the postsynaptic side of asymmetrical synapses and only few synapses exhibited presynaptic KCTD12 labeling along synaptic and extrasynaptic membrane (*Figure 3D*, *Figure 3—figure supplement 2*). In MHb terminals in the rostral IPN, Cav2.3 and KCTD12b showed similar localization patterns, with peak particle densities in the active zone and a gradual decrease in density with increased distance from the active zone (*Figure 3A,E*). In contrast, GABA$_{B1}$, KCTD8, and KCTD12 showed peak localization in the peri-synaptic region with lower particle densities inside the active zone (*Figure 3B–D*). These results suggest that KCTD12b prominently localized to the active zone of ventral MHb terminals in the rostral IPN, whereas KCTD8 dominates the active zone of dorsal MHb terminals in the lateral IPN (*Figure 3F*).

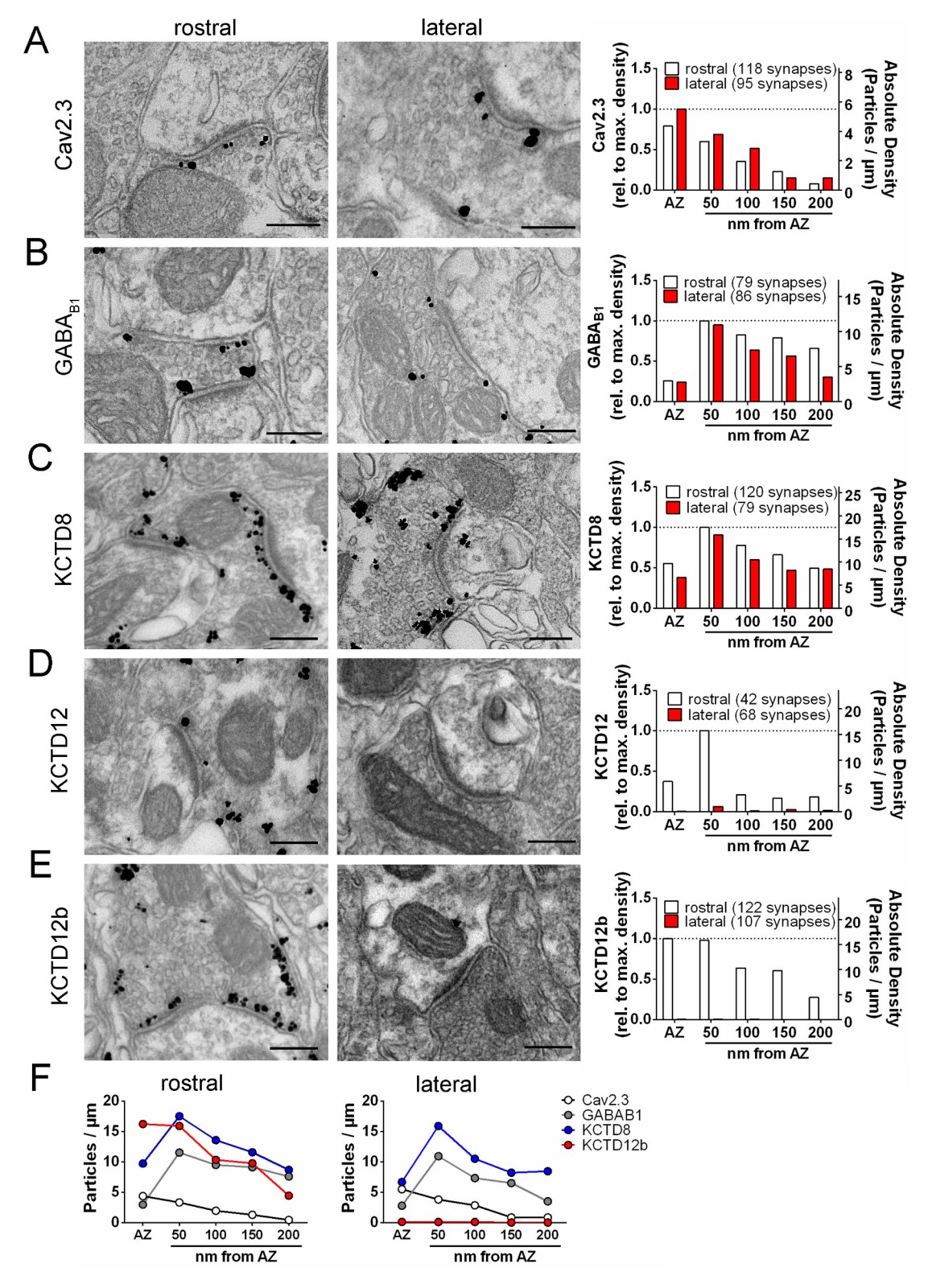

**Figure 3.** Quantification of the presynaptic localization of Cav2.3, GBRs, and KCTDs along the ventral and dorsal MHb-IPN pathways. Transmission electron microscopy images of 70 nm thick sections following pre-embedding immunolabeled IPN slices for Cav2.3 (A), GABA$_{B1}$ (B), KCTD8 (C), KCTD12 (D), and KCTD12b (E) from synapses in the rostral (left images) and lateral (right image) IPN subnuclei. Scale bars: 200 nm. Graph on the right displays quantification of relative and absolute silver-enhanced gold particle densities in the active zone and at distances of 50–200 nm from the edge

*Figure 3 continued on next page*

*Figure 3 continued*

of the active zone (50 nm bins). (**F**) Absolute labeling densities are summarized for synapses in the rostral (left) and lateral IPN (right). Note the absence of KCTD12 and KCTD12b particles in presynaptic terminals inside the lateral IPN subnuclei. KCTD12 was not included in panel F because of predominantly postsynaptic localization inside the rostral IPN. Data was pooled from two animals, showing no significant difference in gold particle distribution patterns with Kolmogorov-Smirnov test (see *Figure 3—figure supplement 1* and *Figure 3—figure supplement 2*).

The online version of this article includes the following source data and figure supplement(s) for figure 3:

**Source data 1.** Quantification of the presynaptic localization of Cav2.3, GBRs, and KCTDs along the ventral and dorsal MHb-IPN pathways.

**Figure supplement 1.** Comparison of distribution of silver-enhanced immunogold particles outside the active zone for Cav2.3, KCTD8, KCTD12, and KCTD12b in MHb terminals inside the rostral and lateral IPN.

**Figure supplement 1—source data 1.** Comparison of distribution of silver-enhanced immunogold particles outside the active zone for Cav2.3, KCTD8, KCTD12, and KCTD12b in MHb terminals inside the rostral and lateral IPN.

**Figure supplement 2.** Two example pictures of postsynaptic immunogold labeling for KCTD12 in the rostral IPN.

To circumvent potential antigen-masking effects due to the protein-dense active zone region in conventional pre-embedding immunolabeling, we performed SDS-digested freeze-fracture replica labeling (SDS-FRL) (*Fujimoto, 1995*). This method enables unhindered access of antibodies to proteins inside/close to the pre- or postsynaptic membrane specialization and allows for multiple labeling with gold particles of distinct sizes (*Hagiwara et al., 2005*; *Indriati et al., 2013*; *Miki et al., 2017*; *Tanaka et al., 2005*). In order to distinguish IPN subnuclei, we used an improved version of the grid-glued SDS-FRL method (*Harada and Shigemoto, 2016*), which facilitates the preservation of complete replicas during the handling procedures, a critical requirement for the identification of IPN subnuclei in the electron microscope (*Figure 4A*). We detected gold particles for Cav2.3 on the P-face of the presynaptic active zone and confirmed antibody specificity using Cav2.3 KO mice (*Figure 4B,C,F*; average density WT: $147.8 \pm 23.9$ particles/$\mu m^2$, n = 4 replicas; KO: $3.4 \pm 0.99$ particles/$\mu m^2$, n = 4 replicas; WT vs. KO: $t_3 = 5.87$, p=0.0099, paired t-test). In addition, we confirmed the concentrated localization of Cav2.3 within the active zone using co-immunolabelings with a mixture of antibodies against marker proteins for the active zone, RIM1/2, neurexin, and CAST (*Figure 4D*; *Miki et al., 2017*). In the absence of marker protein labeling, demarcation of the presynaptic active zone was based on multiple criteria, including P-face curvature and intramembrane particle (IMP) size and density. The area of active zones demarcated without marker protein labeling ($AZ_{unmarked}$) was not significantly different from that with marker labeling ($AZ_{unmarked}$: $0.080 \pm 0.0064$ $\mu m^2$, n = 46 active zones, $AZ_{marked}$: $0.077 \pm 0.0040$ $\mu m^2$, n = 80 active zones; p=0.8692, Kolmogorov–Smirnov test) or from the area of postsynaptic IMP clusters on the E-face, the replica equivalent of the postsynaptic density seen in conventional ultrathin sections (IMP cluster: $0.077 \pm 0.0059$ $\mu m^2$, n = 68 active zones; p>0.9999, Kolmogorov–Smirnov test), verifying our criteria for active zone demarcation.

The result of our pre-embedding immunolabeling suggested that Cav2.3, GBR, and KCTDs are localized in and around the active zone of MHb terminals (*Figure 3*). In order to confirm that these molecules are actually co-localized inside the same terminals, we performed double immunolabelings for Cav2.3 and either $GABA_{B1}$, KCTD8, or KCTD12b in SDS-FRL (*Figure 5*). We focused on these main presynaptic KCTDs because immunogold labeling for KCTD12 was observed in few presynaptic terminals (*Figure 3D*) and was located mostly postsynaptically in the rostral IPN (*Figure 3—figure supplement 2*). We found that in ventral MHb terminals in the rostral IPN, Cav2.3 is co-localized with $GABA_{B1}$, KCTD8, and KCTD12b in over 97% of all Cav2.3-positive active zones (*Figure 5A,B*; $GABA_{B1}$: $98 \pm 0.70\%$, n = 3 replicas; KCTD8: $98 \pm 1.10\%$, n = 3 replicas; KCTD12b: $97 \pm 1.8\%$, n = 4 replicas). Similar co-localization patterns were seen in dorsal MHb terminals located inside the lateral IPN (*Figure 5A,B*; $GABA_{B1}$: $99 \pm 0.69\%$, n = 3 replicas; KCTD8: $97 \pm 0.57\%$, n = 3 replicas), with the exception of an absence of KCTD12b. Particle numbers and densities of all tested molecules, except KCTD12b, were comparable between MHb terminals in the rostral and lateral IPN (*Figure 5B*). In addition, the nearest-neighbor distances (NND) of all tested molecules were significantly smaller than those obtained from simulations of randomly distributed particles (*Figure 5C*), suggesting that Cav2.3, $GABA_{B1}$, and KCTDs are clustered inside the active zone.

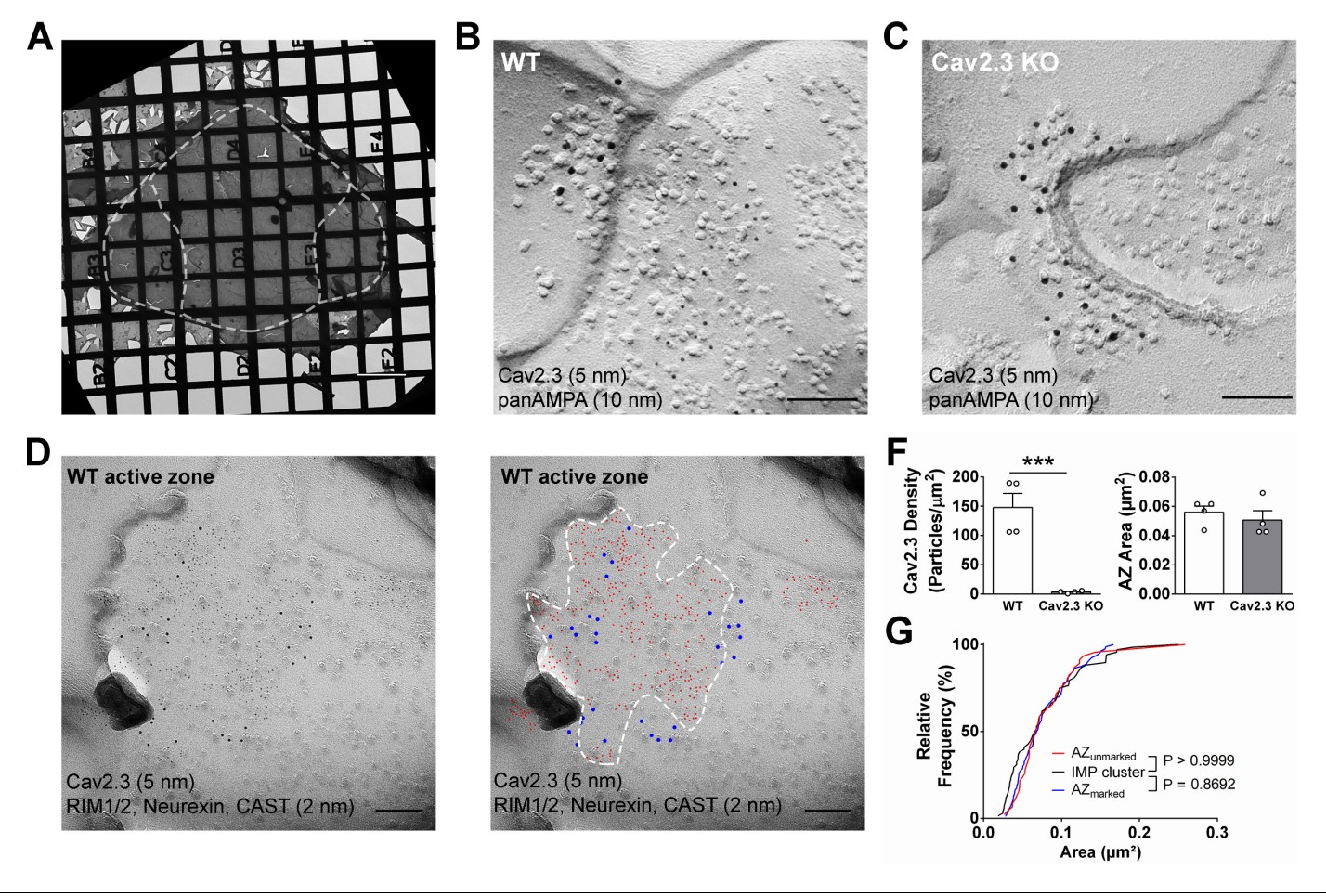

**Figure 4.** SDS-digested freeze-fracture replica labeling confirms Cav2.3 in the active zone of medial habenula terminals in the IPN. (**A**) Example image of a grid-glued replica containing the whole IPN. White dashed line indicates demarcation of rostral/central and lateral subnuclei. Scale bar: 20 µm. (**B**) Example image of a presynaptic P-face and a postsynaptic E-face of a habenular synapse in the rostral IPN that was double labeled with antibodies against AMPA receptors (10 nm gold) and Cav2.3 (5 nm gold). Scale bar: 100 nm. (**C**) Example image of a similar synaptic profile double labeled with antibodies against AMPA receptors (10 nm gold) and Cav2.3 (5 nm gold) in the rostral IPN of a Cav2.3 KO mouse. Scale bar: 100 nm. (**D**) Left: double labeling of a WT carbon-only replica with antibodies against Cav2.3 (5 nm gold) and a mixture of active zone proteins (2 nm gold), including RIM1/2, CAST, and neurexin. Right: the same image with additional coloring of 2 nm (red) and 5 nm (blue) particles and demarcation of the active zone area based on active zone-marker labeling. Scale bars: 100 nm. (**F**) Left: quantification of Cav2.3 labeling densities in the presynaptic P-face in WT and Cav2.3 KO mice. ***p<0.001, unpaired t-test. Right: areas of demarcated active zones, including incomplete profiles, were not significantly different between replicas from WT and Cav2.3 KO mice. Data were obtained from four replicas from four mice of each genotype. (**G**) Comparison of active zone area demarcated with or without active zone markers and the size of the glutamatergic postsynaptic IMP clusters. p-value indicates result of Kolmogorov–Smirnov test.

The online version of this article includes the following source data for figure 4:

**Source data 1.** SDS-digested freeze-fracture replica labeling confirms Cav2.3 in the active zone of medial habenula terminals in the IPN.

## KCTDs modulate release probability at ventral MHb terminals

Based on the exclusive requirement of Cav2.3 for release and the selective expression of KCTD12b co-localized with Cav2.3 in the rostral/central but not lateral IPN, we tested the impact of different KCTDs on neurotransmitter release from ventral MHb terminals in the rostral IPN. We measured the paired-pulse ratio (PPR) of consecutive electrically evoked EPSCs at 20 Hz in WT and KCTD KO mice (*Figure 6A*). PPR was significantly lower in KCTD12b KO mice (1.22 ± 0.11, n = 24 cells) compared with WT (2.40 ± 0.31, n = 27 cells; p=0.0051, Kruskal–Wallis test), KCTD8 KO (2.26 ± 0.27, n = 17 cells; p=0.0080), and KCTD8/12b KO mice (2.76 ± 0.37, n = 11 cells; p=0.0010). In contrast to these changes in basal release, application of 1 µM baclofen still potentiated EPSC amplitudes in all KCTD

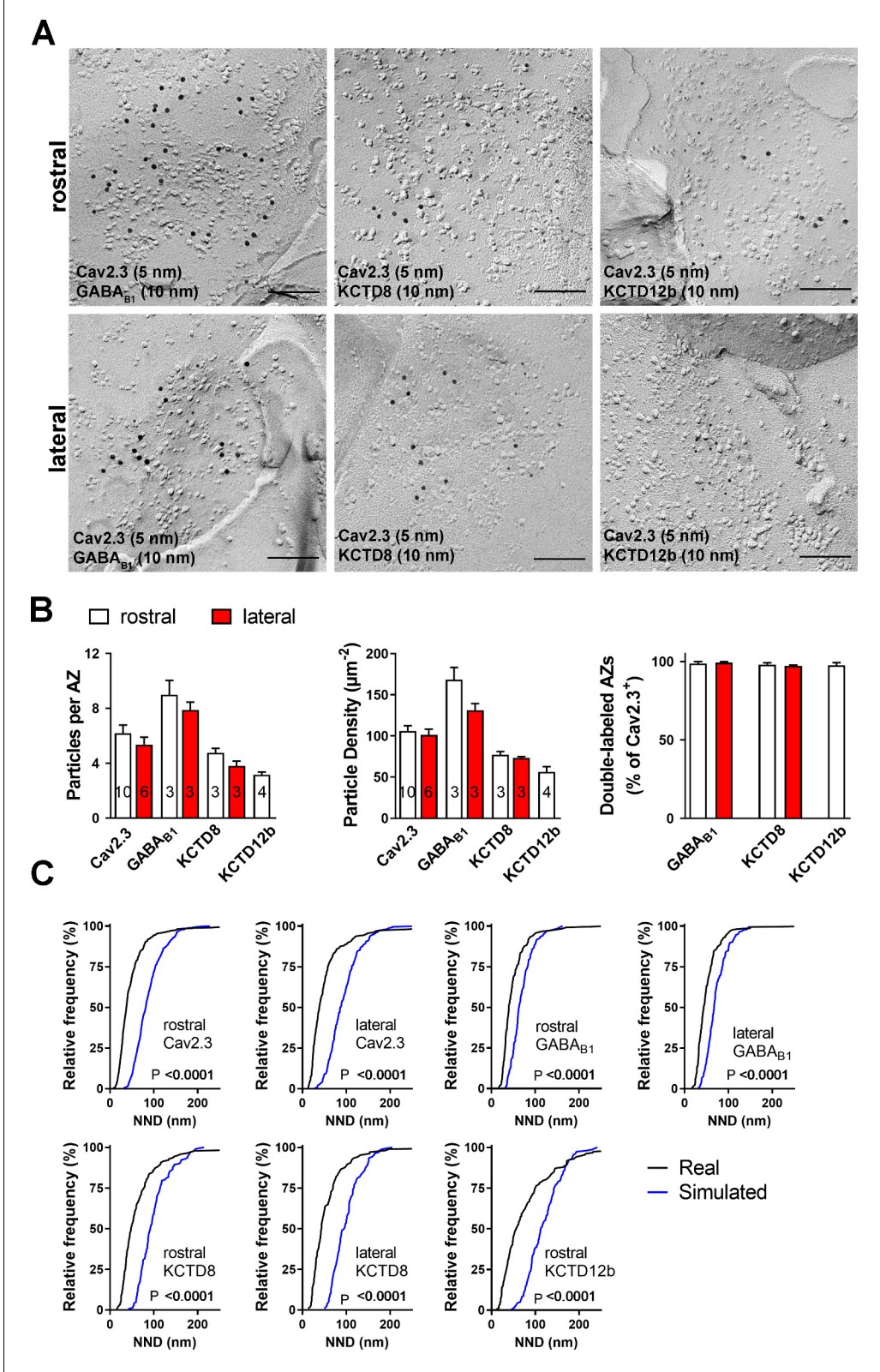

**Figure 5.** Co-localization of Cav2.3 with GBR and KCTDs in the active zone of medial habenula terminals. (**A**) Active zones double labeled for Cav2.3 and either GABA$_{B1}$ (left), KCTD8 (middle), or KCTD12b (right) in IPN replicas. Top row images are from presynaptic terminals in the rostral IPN; bottom row images are from presynaptic terminals in the lateral IPN. Scale bar: 100 nm. (**B**) Quantification of active zone immunolabeling in the rostral and lateral IPN. With the exception of the absence of KCTD12b in lateral IPN terminals, absolute particle numbers per active zone (left graph) and particle

*Figure 5 continued on next page*

*Figure 5 continued*

densities (middle graph) are comparable between MHb terminals in the rostral and lateral IPN. Right graph: Over 97% of active zones positive for Cav2.3 labeling also show labeling for one of the other molecules (GABA$_{B1}$, KCTD8, or KCTD12b), suggesting co-localization of all presynaptic molecules inside the same active zone. Numbers inside the bars indicate the number of replicas used for each quantification. (C) Nearest-neighbor distance (NND) for all presynaptic molecules in MHb terminals inside the rostral and lateral IPN based on the real (black line) and simulated random distribution (blue line). Smaller NND values in real distributions compared to simulation suggest clustering of all presynaptic molecules. p-values calculated via Kolmogorov–Smirnov test.

The online version of this article includes the following source data for figure 5:

**Source data 1.** Co-localization of Cav2.3 with GBR and KCTDs in the active zone of medial habenula terminals.

KO lines with similar intensities (*Figure 6—figure supplement 1*; WT: 615.1 ± 148.9% of baseline; n = 13; KCTD8 KO: 470.8 ± 73.5%; n = 9; KCTD12b KO: 485.4 ± 146.6%; n = 12; KCTD8/12b double KO: 445.8 ± 76.2%; n = 10; $F_{4, 51}$ = 0.4723, p=0.7558, one-way ANOVA), suggesting that KCTDs are not involved in the GBR-mediated enhancement of EPSC amplitudes.

PPR values are generally thought to be inversely correlated with release probability (*Dobrunz and Stevens, 1997*). Therefore, our findings suggest that basal release in KCTD12b KO mice is higher than in WT and KCTD8 KO mice and that this effect is KCTD8 dependent. To confirm the increase in release probability in KCTD12b KO mice, we performed variance-mean analysis (*Figure 6B–E*) to estimate the values of release probability (at 2.5 mM external Ca$^{2+}$) and quantal size (*Figure 6E*; *Clements and Silver, 2000*). Compared with WT mice (WT release probability: 0.26 ± 0.04, n = 16 cells), the release probability from ventral MHb terminals of KCTD12b KO mice (0.49 ± 0.05, n = 9 cells) was significantly increased (*Figure 6E*; $F_{2, 34}$ = 21.23, p<0.0001; WT vs. KCTD12b KO: p=0.0005, one-way ANOVA with Tukey post hoc test). In contrast, the release probability in KCTD8 KO mice (0.12 ± 0.02, n = 12 cells) was significantly lower than that of WT or KCTD12b KO mice (WT vs. KCTD8 KO: p=0.0176; KCTD8 KO vs. KCTD12b KO: p<0.0001; Tukey post hoc test). The quantal size remained unaffected by genotype ($F_{2, 34}$ = 1.613, p=0.2142).

To test the effect of overexpression of KCTD8 on basal release from ventral MHb terminals, we injected lentivirus expressing either EGFP or KCTD8-p2A-EGFP bilaterally into the MHb of WT mice (*Figure 6F*). PPR values recorded in rostral IPN neurons of EGFP control-injected animals remained unchanged (LV-EGFP: 2.17 ± 0.18, n = 17, five mice), whereas KCTD8-injected animals exhibited significantly lower PPR values (LV-KCTD8: 1.41 ± 0.14, n = 13, three mice; $t_{28}$ = 3.20, p=0.0034; unpaired t-test). In contrast, lentiviral expression of KCTD12b in KCTD12b KO mice significantly increased PPR values compared with EGFP control-injected mice (*Figure 6G*; KCTD12b KO EGFP Control: 1.38 ± 0.09, n = 11, three mice; PPR KCTD12b KO rescue: 2.18 ± 0.29, n = 14, three mice; $t_{23}$ = 2.98, p=0.0068, unpaired t-test), resulting in PPR values not significantly different from those of WT mice (WT LV-EGFP vs. KCTD12b rescue: $t_{29}$ = 0.40, p=0.6925, unpaired t-test), confirming that the reduced PPR in KCTD12b KO mice is due to the lack of KCTD12b and not caused by developmental alterations. Overall, these results show that KCTDs differentially modulate basal release probability at ventral MHb terminals. The increase in Cav2.3-mediated release probability by KCTD8 is consistent with its facilitatory effects on Cav2.3 current density in HEK293 cells. The decreased PPR in the absence of KCTD12b recovered to control levels when KCTD8 was additionally absent, suggesting alterations in either Cav2.3 or KCTD8 localization in the active zone of MHb terminals in KCTD12b KO mice. Thus, we next investigated nano-anatomical changes associated with the increased release probability in KCTD12b KO mice.

## Absence of KCTD12b induces a compensatory increase of KCTD8 in the active zone of ventral MHb terminals

To quantify the density of Cav2.3 and KCTD8 in ventral MHb terminals in the rostral IPN, we performed SDS-FRL and co-immunolabeled Cav2.3/GABA$_{B1}$ or Cav2.3/KCTD8 in replicas of WT and KCTD12b KO IPN samples (*Figure 7A*). Since KCTD12b is not expressed in MHb terminals in the lateral IPN, no effects on expression of these molecules are expected in the lateral IPN in KCTD12b KO mice. Thus, we normalized the densities of presynaptic molecules in the rostral IPN to the average density of the same molecule in the corresponding lateral IPN in the same replicas. Thereby, we avoided interference by technical variabilities in labeling efficiencies between individual replicas. We found no significant difference in the densities of Cav2.3 in the presynaptic active zone in the rostral

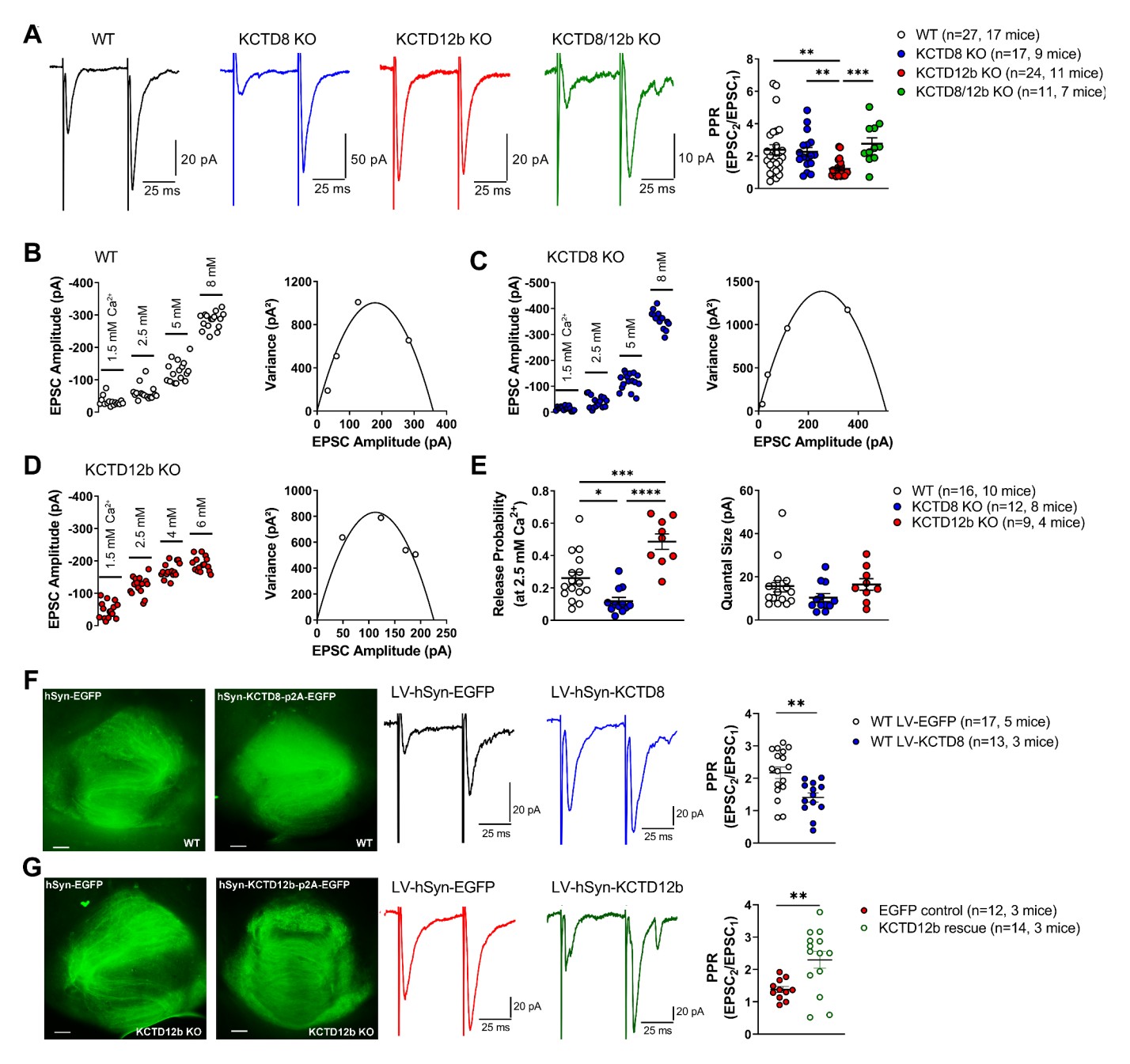

**Figure 6.** KCTDs modulate release probability of ventral MHb terminals. (**A**) In whole-cell recordings from rostral IPN neurons, paired-pulse ratios (PPR) of electrically evoked glutamatergic excitatory postsynaptic currents (EPSCs) were significantly lower in KCTD12b KO mice compared with WT, KCTD8 KO, and KCTD8/12b double KO mice; **p<0.01, ***p<0.001, Kruskal–Wallis with Dunn's post hoc test. (**B–D**) Example variance-mean measurements of electrically evoked EPSC amplitudes at varying external $Ca^{2+}$ concentrations in recorded in rostral IPN neurons of WT (**B**), KCTD8 KO (**C**), and KCTD12b KO mice (**D**). (**E**) Quantification of release probability (left graph) and quantal size (right graph) indicates significantly reduced release probability in KCTD8 KO mice and increased release probability in KCTD12b KO mice. Quantal size remained unaffected. *p<0.05, ***p<0.001, ****p<0.0001 one-way ANOVA with Tukey's post hoc test. (**F**) WT animals were injected with lentivirus expressing either EGFP (WT LV-EGFP) or KCTD8/EGFP (WT LV-KCTD8) into the MHb. Left: Example images of acute IPN slices after electrophysiological recordings showing EGFP signal in fibers inside the IPN. Right: Overexpression of KCTD8 resulted in significantly lower PPR values compared with EGFP controls **p<0.01, unpaired t-test. (**G**) Left: Lentiviral expression of EGFP or KCTD12b/EGFP in acute slices of KCTD12b KO mice. Right: Lentiviral expression of KCTD12b significantly increased PPR values compared with EGFP controls **p<0.01, unpaired t-test. See also *Figure 6—figure supplement 1*.

The online version of this article includes the following source data and figure supplement(s) for figure 6:

**Source data 1.** KCTDs modulate release probability of ventral MHb terminals.

*Figure 6 continued on next page*

*Figure 6 continued*

**Figure supplement 1.** Baclofen still potentiated release in all KCTD KO lines.
**Figure supplement 1—source data 1.** Baclofen still potentiated release in all KCTD KO lines.

relative to the lateral IPN (*Figure 7B*; WT rostral: 1.01 ± 0.04 fold of lateral IPN, n = 8 replicas from eight mice; KCTD12b KO: 1.06 ± 0.13 fold of lateral IPN, n = 8 replicas from eight mice; p=0.5054, Mann–Whitney test). Interestingly, the relative density of KCTD8 in the active zone of MHb terminals was increased approximately twofold in KCTD12b KO mice compared with those of WT mice (*Figure 7B*; WT rostral: 0.84 ± 0.12 fold of lateral IPN, n = 5 replicas from five mice; KCTD12b KO: 2.09 ± 0.45 fold of lateral IPN, n = 5 replicas from five mice; p=0.0079, Mann–Whitney test). This result indicates that the increased release probability in KCTD12b KO mice could be ascribable to a compensatory increase of KCTD8 in the active zone resulting in enhanced association of KCTD8 with Cav2.3. On the other hand, GABA$_{B1}$ expression in the presynaptic active zone of MHb terminals was not significantly different between WT and KCTD12b KO (*Figure 7B*; WT rostral: 1.30 ± 0.18 fold of lateral IPN, n = 3 replicas from three mice; KCTD12b KO: 0.89 ± 0.07 fold of lateral IPN, n = 3 replicas from three mice; p=0.4000, Mann–Whitney test). To confirm the compensatory increase of KCTD8 in ventral MHb terminals in the absence of KCTD12b, we next performed pre-embedding immunolabeling for KCTD8 in presynaptic terminals in rostral and lateral IPN of KCTD12b KO mice (*Figure 7C*). The distribution pattern of KCTD8 in the lateral IPN showed peak densities in the peri-synaptic region, as seen in WT (*Figure 3C*). In contrast, the density of KCTD8 particles in the active zone, but not in the peri- or extrasynaptic area of rostral IPN synapses was significantly increased compared with lateral IPN synapses (main effect of presynaptic distribution: $F_{4, 775}$ = 31.65, p<0.0001; rostral active zone vs. lateral active zone: p<0.0001; two-way ANOVA with Bonferroni post hoc test), showing an altered distribution pattern (compare with *Figure 3C,F*) and similar to that of Cav2.3 and KCTD12b in WT animals (*Figure 3A,E,F*). This observation confirmed that KCTD8 compensates for the loss of KCTD12b in the active zone of ventral MHb terminals.

In order to examine further the spatial proximity of KCTD8 to Cav2.3 in the active zone, we tested whether antibodies bound to KCTD8 might interfere with the labeling for Cav2.3 by steric hindrance in KCTD12b KO mice. We hypothesized that the compensatory increase in KCTD8 in the active zone might lead to increased binding of KCTD8 to Cav2.3 and, therefore, pre-incubation with anti-KCTD8 antibody might reduce Cav2.3 labeling in the active zone. In rostral IPN synapses, we found that the pre-incubation significantly reduced Cav2.3 labeling densities in the active zone, but not the peri- or extrasynaptic regions (*Figure 7D*; main effect of synaptic distribution: $F_{4, 880}$ = 31.42, p<0.0001; control active zone vs. pre-incubation active zone: p=0.0006; two-way ANOVA with Bonferroni post hoc test). In the lateral IPN, pre-incubation slightly but significantly reduced Cav2.3 labeling density only in the peri-synaptic region (*Figure 7E*; main effect of synaptic distribution: $F_{4, 890}$ = 39.01; control <50 nm vs. pre-incubation <50 nm: p=0.0320). The peri-synaptic region is the location exhibiting peak expression for KCTD8 in dorsal MHb terminals. The significant reduction in Cav2.3 labeling density at this position therefore validates our steric hindrance approach. These results support that KCTDs in the active zone dynamically regulate the release probability, which may be important for scaling synaptic strength independent from GBRs in the MHb-IPN pathway.

In summary, the active zone of ventral MHb terminals predominantly contains Cav2.3 and KCTD12b, whereas the peri-synaptic region exhibits high levels of KCTD8 and GBRs (*Figure 8*, WT terminal). In the absence of KCTD12b, KCTD8 enters the active zone to interact with Cav2.3, thereby increasing Ca$^{2+}$ influx and ultimately release (*Figure 8*, KCTD12b KO terminal).

## Discussion

We used a combination of anatomical, biochemical, and functional methods to study the role of KCTDs at two parallel pathways from the MHb to the IPN and identified an unexpected function of KCTDs in the modulation of neurotransmitter release. We confirmed that Cav2.3 is involved in mediating transmitter release at both pathways, but only release at the ventral MHb-IPN pathway exclusively relies on Cav2.3. KCTD8 and KCTD12b show distinct expression along these pathways, with KCTD12b specifically expressed in the cholinergic ventral MHb-IPN pathway. In HEK293T cells, we

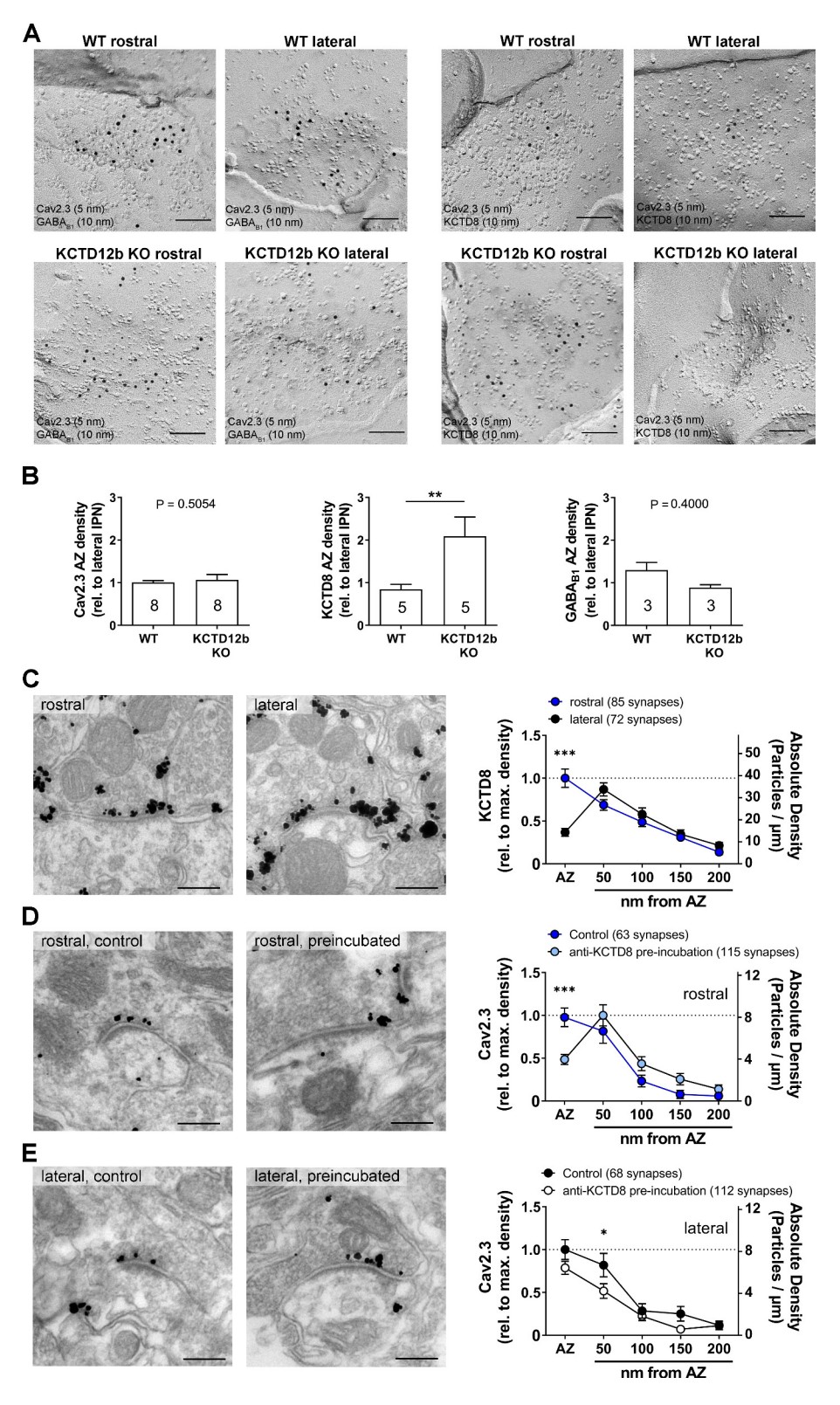

**Figure 7.** Absence of KCTD12b leads to a compensatory increase of KCTD8 inside the active zone of ventral MHb terminals. (**A**) Example images of active zones containing Cav2.3 and either GABA$_{B1}$ (left) or KCTD8 (right) in replicas of WT (upper row) and KCTD12b KO IPN tissue (lower row). Scale bars: 100 nm. (**B**) Quantification of relative densities for Cav2.3, KCTD8, and GABA$_{B1}$ in active zones located in the rostral IPN of WT and KCTD12b KO mice. Densities were normalized to the average density in MHb terminals inside the lateral IPN of the same replica. The number inside the bars

*Figure 7 continued on next page*

*Figure 7 continued*

indicates the number of replicas used for quantification. **$p<0.01$ in a Mann–Whitney test. (C) Pre-embedding EM labeling for KCTD8 in KCTD12b KO mice (data from three mice). The density of KCTD8-labeled gold particles was significantly higher in the AZ of ventral MHb terminals in the rostral IPN compared with dorsal MHb terminals in the lateral IPN. ***$p<0.0001$ two-way ANOVA with Bonferroni post hoc test. (D, E) Comparison of pre-embedding EM labeling for Cav2.3 in MHb terminals in KCTD12b KO mice of control sections (standard labeling, four mice) with sections pre-incubated with anti-KCTD8 primary antibody and biotinylated secondary antibody (three mice). Pre-incubation resulted in significant reduction in labeling densities for Cav2.3 selectively in the active zone of ventral MHb terminals in the rostral IPN as well as in the peri-synaptic region of dorsal MHb terminals in the lateral IPN. ***$p<0.001$, *$p<0.05$ two-way ANOVA with Bonferroni post hoc test.

The online version of this article includes the following source data for figure 7:

**Source data 1.** Absence of KCTD12b leads to a compensatory increase of KCTD8 inside the active zone of ventral MHb terminals.

found that KCTD8 and KCTD12b bind to Cav2.3 at the plasma membrane and that specifically KCTD8 increases Cav2.3 currents. Using electron microscopy, we verified that Cav2.3 and KCTDs co-localize within the presynaptic active zone of MHb terminals. Genetic deletion of the KCTDs indicated that KCTD8 increases basal release probability and that KCTD12b prevents this effect. Synapses lacking KCTD12b showed an increased density of KCTD8 in the active zone, resulting in a close association of KCTD8 with Cav2.3 and a consequent increase in release probability. Viral expression of KCTD12b in KCTD12b KO mice reduced release probability to WT levels, whereas overexpression of KCTD8 in WT mice mimicked the phenotype of KCTD12b KO mice. Our results support that the KCTDs scale synaptic strength at the MHb-IPN pathway independent of GBRs.

## Dominant presynaptic localization and function of Cav2.3 in MHb-IPN pathway

In contrast to other brain areas, R-type $Ca^{2+}$ channels may be the exclusive source of presynaptic $Ca^{2+}$ influx to trigger release from ventral MHb terminals, given that neither P/Q- nor N-type $Ca^{2+}$ channels are expressed in MHb neurons (*Ludwig et al., 1997*). The finding by *Ludwig et al., 1997*

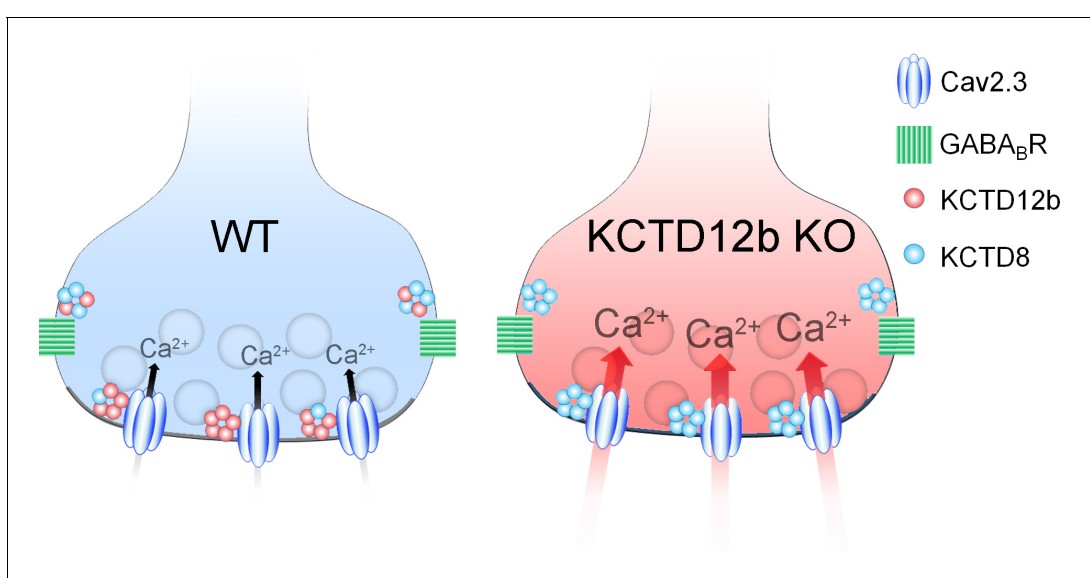

**Figure 8.** Summary of presynaptic localization of Cav2.3, KCTDs, and GBRs in ventral MHb neurons of WT and KCTD12b KO mice. Schematic representation of the distribution and function of Cav2.3, GBRs, and KCTDs in ventral MHb terminals of WT and KCTD12b KO mice. Left: In WT terminals, the active zone contains Cav2.3 and hetero-pentameric rings comprising KCTD12b in excess over KCTD8, whereas KCTD8 and GBRs are located peri-synaptically. Right: In absence of KCTD12b, KCTD8 invades the active zone and compensates for the loss of KCTD12b, resulting in increased release probability, potentially via increased $Ca^{2+}$ influx. See *Figure 8—figure supplement 1*.

The online version of this article includes the following figure supplement(s) for figure 8:

**Figure supplement 1.** Allen Brain Atlas images showing lack of expression of Cav2.1 (http://mouse.brain-map.org/gene/show/12071) and Cav2.2 (http://mouse.brain-map.org/gene/show/12072) mRNA in MHb neurons, whereas Cav2.3 shows strong expression in MHb neurons (http://mouse.brain-map.org/gene/show/12075).

fits the data of the Allen Brain Atlas, which shows no expression of Cav2.1 and Cav2.2 in MHb neurons (*Figure 8—figure supplement 1*; images from Allen Institute for Brain Science, 2004) (*Lein et al., 2007*). Although the predominant presynaptic localization of Cav2.3 immunoreactivity was previously reported (*Parajuli et al., 2012*), the functional implications of this localization were not determined. Our data confirmed that both MHb-IPN pathways heavily rely on Cav2.3 for release. Interestingly, projections from the ventral MHb to the rostral IPN were more affected by SNX-482 than the dorsal MHb to lateral IPN pathway. Using Cav2.3 KO mice, we found residual EPSC responses in lateral but not rostral IPN neurons, confirming that exclusively Cav2.3 mediates transmitter release from ventral MHb terminals. The remaining release from dorsal MHb terminals in the lateral IPN of Cav2.3 KO mice could be explained either by a compensatory upregulation of other $Ca^{2+}$ channels or by the presence of additional $Ca^{2+}$ channels in WT mice. In dorsal MHb terminals of WT mice, the SNX-482-insensitive EPSC component was also insensitive to ω-conotoxin GVIA, ruling out N-type $Ca^{2+}$ channels, but was completely abolished by $CdCl_2$. Therefore, dorsal MHb terminals likely express Cav2.3 together with other $Ca^{2+}$ channels, potentially L-type $Ca^{2+}$ channels.

## Role of KCTDs in potentiation of Cav2.3-mediated release by GBR activation

A previous report of the potentiating action of presynaptic GBRs on cholinergic MHb terminals identified Cav2.3 as a critical mediator of this effect, based on the observations that both genetic ablation of Cav2.3 and pharmacological inhibition of Cav2.3 with $Ni^+$ prevented the potentiation (*Zhang et al., 2016*). Given the modulatory roles of KCTDs in shaping GBR effector responses (*Fritzius et al., 2017*), we initially hypothesized that the unique repertoire of KCTDs in ventral MHb neurons may be involved in the facilitatory action of GBRs. However, deletion of neither KCTD8, KCTD12b, nor both prevented the potentiation of EPSC amplitudes by baclofen, suggesting a KCTD-independent effect.

## Regulation of Cav2.3-mediated release by KCTDs

In addition to the co-localization of Cav2.3 with KCTDs at the active zone in habenular terminals, we identified a hitherto unknown direct interaction of Cav2.3 with KCTD8 and KCTD12b in vitro. Previous proteomics studies revealed co-precipitations of N-type/Cav2.2 $Ca^{2+}$ channels with KCTD8 and KCTD16, but no other KCTD−Cav2 interactions were reported (*Müller et al., 2010*; *Schwenk et al., 2016*). Possibly, the use of total brain extracts may have limited detection to the most prominent protein interactions. Thus, brain-wide approaches to study presynaptic $Ca^{2+}$ channel-interacting proteins may have failed to detect interactions uniquely occurring inside MHb terminals. In contrast, our pre-embedding EM approach utilizing interference in Cav2.3 immunogold labeling by steric hindrance with KCTD8 antibody in the AZ of MHb terminals (*Figure 7D*) indicates a close association in situ of these molecules within 16 nm (two times of IgG size 8 nm), if not a direct binding.

The selective KO of KCTD8 lowered basal levels of release probability from ventral MHb terminals below that of WT synapses. On the other hand, KO of KCTD12b resulted in a significant increase of basal release probabilities. This finding of modulation of release independent of GBR activation could be explained by the current-enhancing effects of KCTD8 on Cav2.3 observed in a heterologous expression system. Since the probability of neurotransmitter release is relatively low in WT terminals (0.26), the additional reduction in release probability in KCTD8 KO mice (0.12) may not be detectable by PPR measures. An important indication that KCTD8 functionally increases synaptic strength was the observation that viral overexpression of KCTD8 in WT ventral MHb neurons significantly lowered PPR values compared with EGFP controls. This result suggests that KCTD8 and KCTD12b may be in a state of dynamic balance. Creating an imbalance appears to result in alterations in synaptic strength. This possibility would be in line with our pre-embedding EM result, with KCTD12b showing peak densities inside the active zone of ventral MHb terminals, whereas KCTD8 shows peak expression in the peri-synaptic region. This could be an indication for the preferred interaction of synaptic Cav2.3 with KCTD12b over KCTD8 due to spatial proximity or higher binding affinity. Although the increase in release probability at ventral MHb terminals is likely caused by a direct effect of KCTD8 on Cav2.3, it is unknown whether other $Ca^{2+}$ channels involved in transmitter release from dorsal MHb terminals are equally affected by KCTD8.

It is unclear whether a compensatory increase of KCTD12b occurs in KCTD8 KO mice. Our initial results from WT pre-embedding EM indicate that KCTD12b displays peak densities in the active zone, whereas KCTD8 shows peak densities in the peri-synaptic region. Therefore, although it may be possible to see a compensatory increase in KCTD12b in the active zone of KCTD8 KO mice, due to the weaker expression of KCTD8 in the WT active zone, such an increase would be expected to be smaller than the compensatory increase seen in KCTD12b KO animals. Furthermore, the functional consequence of a compensatory increase in KCTD12b in the active zone in KCTD8 KO mice would not be detected in our functional experiments, as KCTD12b did not alter Cav2.3 function in vitro and KCTD12b has no effects on PPR in the absence of KCTD8 (no significant difference between KCTD8 KO and KCTD8/12b double KO in *Figure 6A*). Our results suggest that the relatively mild reduction in release probability seen in KCTD8 KO mice resulted from removal of the comparatively low levels of KCTD8 from the WT active zone, rather than from a compensatory increase in KCTD12b.

To induce synaptic plasticity in the ventral MHb to IPN pathway, release of glutamate, activation of postsynaptic $Ca^{2+}$-permeable AMPA receptors on GABAergic IPN neurons, and subsequent retrograde release of GABA to activate presynaptic $GABA_B$ receptors on MHb terminals are required (*Koppensteiner et al., 2017*). Therefore, KCTD-mediated modulation of release probability could determine a ventral MHb synapse's propensity to undergo and express activity-dependent plasticity. Future studies may provide additional insights into whether physiological learning paradigms affecting the MHb-IPN pathway, such as the formation and extinction of aversive memories (*Agetsuma et al., 2010*; *Koppensteiner et al., 2017*; *Melani et al., 2019*; *Zhang et al., 2016*), could alter the ratio of KCTD8 to 12b.

Overall, our study provided new insights into the physiological role of presynaptic Cav2.3, GBRs, and their auxiliary KCTD subunits in an evolutionary conserved neuronal circuit. It remains to be determined whether the prominent presence of presynaptic KCTDs at other synapses (*Müller et al., 2010*) could be an indication of similar neuromodulatory function of KCTDs in different pathways of the brain.

# Materials and methods

## Key resources table

| Reagent type (species) or resource | Designation | Source or reference | Identifiers | Additional information |
|---|---|---|---|---|
| Strain, strain background (*Mus musculus*, ♂) | C57BL/6J | The Jackson Laboratory | #000664 | |
| Strain, strain background (*Mus musculus*, ♂) | BALB/cJ | The Jackson Laboratory | #000651 | |
| Strain, strain background (*Mus musculus*, ♂) | Cav2.3 KO (gene: *Cacna1e*) | Prof. Tsutomu Tanabe, (*Saegusa et al., 2000*) | | |
| Strain, strain background (*Mus musculus*, ♂) | KCTD8 KO (gene: *Kctd8*) | Prof. Bernhard Bettler, (*Schwenk et al., 2010*) | | |
| Strain, strain background (*Mus musculus*, ♂) | KCTD12 KO (gene: *Kctd12*) | Prof. Bernhard Bettler, (*Schwenk et al., 2010*) | | |
| Strain, strain background (*Mus musculus*, ♂) | KCTD12b KO (gene: *Kctd12b*) | Prof. Bernhard Bettler, (*Schwenk et al., 2010*) | | |
| Strain, strain background (*Mus musculus*, ♂) | KCTD8/12b double KO (genes: *Kctd8/Kctd12* b) | This paper | | |

*Continued on next page*

*Continued*

| Reagent type (species) or resource | Designation | Source or reference | Identifiers | Additional information |
|---|---|---|---|---|
| Strain, strain background (*Mus musculus*, ♂♀) | Tac1-IRES-Cre (B6;129S-Tac1$^{tm1.1(cre)Hze}$/J) | The Jackson Laboratory | #021877 | |
| Strain, strain background (*Mus musculus*, ♂♀) | Ai32 (B6;129S-Gt(ROSA) 26 Sor$^{tm32(CAG-COP4*H134R/EYFP)Hze}$/J) | The Jackson Laboratory | #012569 | |
| Strain, strain background (*Mus musculus*, ♂) | Tac1-ChR2-EYFP | Offspring of Tac1-IRES-Cre and Ai32 mice | | |
| Genetic reagent (virus) | LV–hSyn–3xFlag/mKCTD8[NM_175519.5]/P2A/EGFP | VectorBuilder GmbH | | Custom-made |
| Genetic reagent (virus) | LV–hSyn–3xFlag/mKCTD12b[NM_175429.4]/P2A/EGFP | VectorBuilder GmbH | | Custom-made |
| Genetic reagent (virus) | LV–hSyn–EGFP | VectorBuilder GmbH | | Custom-made |
| Cell line (human) | HEK293T | ATCC, (*Seddik et al., 2012*) | https://web.expasy.org/cellosaurus/CVCL_0063 | |
| Cell line (human) | HEK293 cells stably expressing α1E-3 | Prof. David J. Adams, (*Berecki et al., 2014*) | | |
| Antibody | Guinea pig polyclonal anti-Cav2.3 | Genovac, (*Parajuli et al., 2012*) | | Custom-made antibody, 1 µg/ml for IHC, 8 µg/ml for SDS-FRL |
| Antibody | Rabbit polyclonal anti-GABAB1 | Prof. Akos Kulik, (*Kulik et al., 2002*) | | Custom-made antibody, final concentration: (1 µg/ml) for IHC, (2 µg/ml) for SDS-FRL |
| Antibody | rabbit polyclonal anti-KCTD8 | Prof. Bernhard Bettler, (*Schwenk et al., 2010*) | | Custom-made antibody, final concentration: (1 µg/ml) for IHC, (4 µg/ml) for SDS-FRL |
| Antibody | Rabbit polyclonal anti-KCTD12 | Prof. Bernhard Bettler, (*Schwenk et al., 2010*) | | Custom-made antibody, final concentration: (1 µg/ml) for IHC, (4 µg/ml) for SDS-FRL |
| Antibody | Rabbit polyclonal anti-KCTD12b | Prof. Bernhard Bettler, (*Schwenk et al., 2010*) | | Custom-made antibody, final concentration: (1 µg/ml) for IHC, (4 µg/ml) for SDS-FRL |
| Antibody | Rabbit polyclonal anti-RIM1/2 | Synaptic Systems | 140 203 | final concentration: (5 µg/ml) for SDS-FRL |
| Antibody | Rabbit polyclonal anti-CAST | Prof. Watanabe, (*Hagiwara et al., 2018*) | | final concentration: (3 µg/ml) for SDS-FRL |
| Antibody | Rabbit polyclonal anti-Neurexin | Prof. Watanabe, (*Miki et al., 2017*) | | final concentration: (5 µg/ml) for SDS-FRL |
| Peptide, recombinant protein | SNX-482 | hellobio | HB1235 | |

*Continued on next page*

*Continued*

| Reagent type (species) or resource | Designation | Source or reference | Identifiers | Additional information |
|---|---|---|---|---|
| Peptide, recombinant protein | ω-Conotoxin GVIA | Alomone Labs | 106375-28-4 | |
| commercial assay or kit | Plasma membrane extraction kit | Abcam | ab 65400 | |
| Chemical compound, drug | R(+)-Baclofen hydrochloride | Merck | G013 | |
| Chemical compound, drug | 1(S),9(R)-(—)-Bicucullin-Methiodid | Merck | 14343 | |
| Chemical compound, drug | Hexamethonium bromide | Tocris | 4111 | |
| Chemical compound, drug | Mecamylamine hydrochloride | Tocris | 2843 | |
| Chemical compound, drug | Cadmiumchloride | Merck | 202908 | |
| Software, algorithm | Graphpad Prism 8 | Graphpad | https://www.graphpad.com/scientific-software/prism/ | |
| Software, algorithm | MATLAB | MathWorks | https://www.mathworks.com/products/matlab.html?s_tid=hp_products_matlab | |
| Software, algorithm | Reconstruct | John C. Fiala, Ph.D. | https://synapseweb.clm.utexas.edu/software-0 | |

## Animals

Wild-type C57BL/6J (Jax, Bar Harbor, ME; #000664) and BALB/cJ (Jax, #000651) mice were initially purchased from Jackson Laboratory. Homozygous KCTD KO lines were generated by the lab of Bernhard Bettler at the University of Basel (*Schwenk et al., 2010*). For KCTD8 KO line generation, exon one containing ATG and most of the open reading frame (ORF) of the Kctd8 gene (MGI:2443804) was replaced with a loxP-flanked neo. To generate KCTD12 KO mice, 5′ part of exon one containing complete ORF of the Kctd12 gene (MGI:2145823) was replaced with a loxP-flanked neo (*Cathomas et al., 2015*). Similarly, KCTD12b KO line was generated by replacing the 5′ part of exon three containing complete ORF of the Kctd12b gene (MGI:2444667) with a loxP-flanked neo. All KCTD KO lines had the neo removed by crossing founders with a Cre-deleter line, leaving one loxP site behind. The KCTD8/12d double KO line was generated by mating F2 hybrids of the parental lines. Background strains of KCTD KO lines were as follows: KCTD8 (C57BL/6J and 129), KCTD12 (C57BL/6J, 129, BALB/cJ), KCTD12b (BALB/cJ), and KCTD8/12b (C57BL/6J, 129, BALB/cJ). Cav2.3 KO mice were generated by the lab of Tsutomu Tanabe (*Saegusa et al., 2000*). To obtain Tachykinin1 (Tac1)-ChR2-EYFP mice, we crossed Tac1-Cre (Jax, #021877) with Ai32 (Jax, #012569) mice. All mice were bred at the preclinical facility of IST Austria on a 12:12 light–dark cycle with access to food and water ad libitum. All experiments were performed in accordance with the license approved by the Austrian Federal Ministry of Science and Research (Animal license number: BMWFW-66.018/0012-WF/V/3b/2016) and the Austrian and EU animal laws. Only male mice aged 2–5 months were used for all experiments.

## Transcardial perfusion for brain fixation

Mice were anaesthetized with a mixture of ketamine (90 mg/kg body weight) xylazine (4.5 mg/kg) solution intraperitoneally, and 25 mM ice-cold phosphate-buffered saline (PBS) was transcardially perfused through the left ventricle at a flow rate of 7 ml/min for 30–60 s. Subsequently, paraformaldehyde (PFA) solution was perfused for 12 min. PFA solutions of different concentrations were used for confocal imaging (4% PFA [TAAB Laboratories Equipment Ltd., Aldermaston, UK] and 15% picric acid in 0.1 M phosphate buffer, PB), pre-embedding (4% PFA and 15% picric acid in 0.1 M PB +0.05% glutaraldehyde [TAAB]), and SDS-digested freeze-fracture replica labeling (SDS-FRL, 2% PFA and 15% picric acid in 0.1 M PB). The pH of all PFA solutions was adjusted to 7.4 ± 0.05 with HCl. After perfusion, the brain was excised and placed in 0.1 M PB and sectioned within 3 days. Slices of different thickness (50 µm for confocal microscopy and pre-embedding, 70 µm for SDS-FRL) were cut with a vibratome (Linear-Pro7, Dosaka, Japan) in ice-cold 0.1 M PB.

## Immunohistochemistry

Brain slices containing the IPN were washed with phosphate-buffered saline (PBS) and subsequently incubated in blocking buffer (10% normal goat serum, 2% BSA, 0.5% Triton-X100 in 0.1 M PBS) for 1 hr prior to incubation with primary antibodies: guinea pig anti-Cav2.3 (1 µg/ml, two overnight [O/N], Genovac), rabbit anti-GABAB1 (B17, 1 µg/µl, 1 O/N [*Kulik et al., 2002*]), rabbit anti-KCTD8 (1 µg/µl, Bettler lab, Univ. Basel), rabbit anti-KCTD12 (1 µg/µl, 1 O/N, Bettler lab, Univ. Basel), and rabbit anti-KCTD12b (polyclonal, raised against a synthetic peptide comprised of the N-terminal amino acids 1–16 of KCTD12b) (1 µg/µl, 1 O/N, Bettler lab, Univ. Basel) (*Metz et al., 2011*; *Schwenk et al., 2010*). Brain sections were washed in PBS and subsequently incubated for 1 hr in secondary antibody (1:500, Alexa-488 anti-guinea pig [Molecular Probes, Eugene, OR] or Alexa-488 anti-rabbit [Molecular Probes]). Sections were mounted onto glass slides, and images were taken with an LSM 800 (Zeiss, Oberkochen, Germany) confocal microscope.

## Pre-embedding immunolabeling

Brain slices were washed in 0.1 M PB (two times, 10 min each) and cryo-protected by incubation in 0.1 M PB containing 20% sucrose O/N at 4°C. The next day, slices underwent three cycles of freeze-thawing by freezing the slices on liquid nitrogen for 1 min and thawing them in 20% sucrose on a hot plate (50°C) for 2 min. Slices were washed in 0.1 M PB (10 min) followed by washing in TBS (three times, 20 min each). Free aldehydes were quenched by incubating slices in 50 mM glycine (Sigma–Aldrich, St. Louis, MO) in TBS (10 min). After washing in TBS (three times, 20 min each), slices were blocked with blocking buffer (10% NGS, 2% BSA in TBS, 1 hr). Primary antibody incubation was done with respective antibodies in 2% BSA solution for 2 O/N at 4°C. The concentration of antibodies was 1 µg/ml for Cav2.3, KCTD8, KCTD12, KCTD12b, and GABA$_{B1}$. For steric hindrance experiments, slices were pre-incubated with 2 µg/ml of anti-KCTD8 antibody (2 O/N), followed by incubation with biotinylated anti-rabbit secondary antibody (1:100, 2 O/N) before incubation with anti-Cav2.3 antibody (2 O/N). Subsequently, slices were rinsed in TBS (three times, 20 min each) and incubated in respective secondary antibodies (1:100) O/N at 4°C in 2% BSA in TBS. For Cav2.3, 1.4 nm gold-conjugated anti-guinea pig antibody (Nanoprobes, Yaphank, NY) and for all other antibodies, i.e. GABAB1, KCTD8, KCTD12, and KCTD12b, 1.4 nm gold-conjugated anti-rabbit antibody (Nanoprobes) was used. Slices were washed in TBS and PBS (two times, 20 min each) followed by post-fixation in 1% glutaraldehyde in PBS (10 min), washing in PBS (three times, 10 min), and quenching of free glutaraldehyde in 50 mM glycine in PBS (10 min). Finally, slices were washed in PBS (three times, 10 min each) and Milli-Q (MQ) H$_2$O (three times, 5 min each).

For silver intensification of immunogold particles, Nanoprobes silver intensification (Nanoprobes) component A (initiator) and B (moderator) were mixed and vortexed, followed by the addition of component C (activator). After vortexing, slices were incubated in the mixture for 9 min 15 s in the dark. Tubes were tapped every 2 min for uniform intensification. Slices were washed with MQ water (three times, 10 min each) and in 0.1 M PB (10 min), followed by post-fixation in 1% OsO$_4$ in 0.1 M PB (20 min in the dark). After osmification, slices were washed in 0.1 M PB (10 min) and in MQ water (three times, 5 min each) and counter-stained in 1% uranyl acetate (Al-labortechnik, Zeillern, Germany) in MQ H$_2$O (35 min in the dark). Subsequently, slices were serially dehydrated in ethanol solutions of different concentrations in ascending order up to 100% (50–95% ethanol in five steps, 5 min

each; 100% ethanol two times, 10 min each) and then washed with propylene oxide (Sigma–Aldrich; two times, 10 min each). Slices were then submerged in Durcupan resin (Sigma –Aldrich; mixture of components A, B, C, and D in proportion of 10:10:0.3:0.3, respectively) for 1 O/N at room temperature (RT).

For flat embedding, each slice was isolated on a silicon-coated glass slide, covered with an ACLAR fluoropolymer film (Science Services, Munich, Germany) and incubated at 37°C (1 hr) followed by incubation at 60°C (2 O/N). For re-embedding, tissue containing the rostral or lateral IPN was excised with a surgical blade, placed into the lid of a plastic tube (TAAB), which was then filled with Durcupan resin and incubated at 60°C (2 O/N). Each resin block was trimmed using a Leica EM TRIM2 to remove the resin surrounding the sample. The resin above the sample in the trimmed block was further cut at 200 nm increments using a glass knife in the ultramicrotome Leica EM UC7 until the sample was exposed. Seventy nanometer sections were cut with a diamond knife (Diatome Ultra 45˚). A small ribbon of floating sections was collected and mounted onto a copper-grid coated with formvar. Once the grid was dry, it was stored in a grid box for further observation in Tecnai10 (FEI; accelerating voltage 80 kV) or Tecnai 12 (FEI; accelerating voltage 120 kV) transmission electron microscopes.

## SDS-digested freeze-fracture replica preparation and labeling

Brain slices (70 µm) of mice transcardially perfused with 2% PFA in 0.1 M PB were prepared, and the whole IPN was manually excised. Tissue was then incubated in 30% glycerol in 0.1 M PB O/N for cryo-protection. The next day, tissue samples were transferred into gold or copper carriers and frozen under high pressure (>300 bar) using an HPM010 (Leica, Wetzlar, Germany). Frozen samples were stored in liquid nitrogen until further processing. To craft freeze-fracture replicas, two gold carriers containing frozen tissue samples were placed in a carrier holder in liquid nitrogen, which was inserted into the freeze-fracture machine (BAF060, Leica) and left to equilibrate to –117°C under high vacuum ($2.0 \times 10^{-7}$–$1.0 \times 10^{-6}$ mbar) for 20 min. Subsequently, tissue was fractured and a carbon layer (5 nm at a rate of 0.1–0.3 nm/s) was evaporated onto the tissue at a 90˚ angle, followed by a platinum/carbon layer (2 nm at a rate of 0.06–0.1 nm/s) applied at a 60˚ angle and another carbon layer (20 nm at a rate of 0.3–0.6 nm/s) applied at a 90˚ angle. For the preparation of carbon-only replicas, the second layer consisted of a 5 nm carbon layer applied at a 60˚ angle. After evaporation, replicas were removed from the machine and placed in tris-buffered saline (TBS, 50 mM). Subsequently, replicas were glued (tissue-side up) onto a finder grid and the glue (optical adhesive 61, Norland, Cranbury, NJ) was hardened by applying UV light for 20 s and then transferred into SDS-solution containing: 2.5% SDS, 20% sucrose in 15 mM Tris buffer (pH 8.3). Tissue was subsequently digested by incubating the replicas for 48 hr at 60°C under gentle agitation (50 rpm shaker), followed by incubation for 12–15 hr at 37°C under gentle agitation.

For immunolabeling of SDS-digested replicas, replicas were washed in washing buffer (containing 0.1% Tween-20, 0.05% bovine serum albumin [BSA], 0.05% $NaN_3$pH = 7.4) and incubated in blocking buffer (washing buffer+ 5% BSA) for 1 hr. Replicas were transferred to blocking buffer containing primary antibody (ginea pig anti-Cav2.3, 8 µg/µl) followed by incubation at 15°C O/N. Thereafter, replicas were washed and incubated in blocking solution containing secondary antibodies (2 nm gold-conjugated anti-rabbit, 5 nm gold-conjugated anti-ginea pig, or 10 nm gold-conjugated anti-rabbit, all diluted 1:30) at 15°C O/N. The following day, the antibody labeling procedure was repeated for the next primary antibody (4 µg/ml for anti-KCTD8, anti-KCTD12, anti-KCTD12b; 2 µg/ml for anti-GABAB1) and corresponding secondary antibodies. Labeled grid-glued replicas received a final carbon layer (20 nm) onto the labeled replica side using a High-Vacuum Coater ACE600 (Leica), followed by dissolution of the glue in Dynasolve 711 (Dynaloy, Indianapolis, IN) at 60°C under gentle agitation (60 rpm) for 2 hr. Solvent was subsequently removed by washing the grid in 100% methanol for 10 min followed by ethanol (100, 95, 90, 70, 50%, 5 min each). After short air drying, replicas were stored in grid boxes until observation under the transmission electron microscope.

## Acute brain slice electrophysiology

Mice were anesthetized with a mixture of ketamine (90 mg/kg) and xylazine (4.5 mg/kg) and transcardially perfused with ice-cold, oxygenated (95% $O_2$, 5% $CO_2$) artificial cerebrospinal fluid (ACSF) containing (in mM) 118 NaCl, 2.5 KCl, 1.5 $MgSO_4$, 1 $CaCl_2$, 1.25 $NaH_2PO_4$, 10 D-glucose, 30

NaHCO$_3$, (pH = 7.4). The brain was rapidly excised, and coronal brain slices of 250 μm thickness were prepared with a Dosaka Linear-Pro7. For SNX-482 experiments in rostral IPN, brain slices were prepared at a 54° angle to allow for electrical stimulation of FR. For angled slice recordings in lateral IPN of WT and Cav2.3 KO mice (*Figure 1H–K*), slices were cut at 1 mm thickness to assure that FR remained intact all the way to the lateral IPN. Slices were recovered at 35°C for 20 min and thereafter slowly cooled down to RT over the course of 1 hr. After recovery, one slice was transferred to the recording chamber (RC-26GLP, Warner Instruments, Holliston, MA) and superfused with ACSF containing 2.5 mM CaCl$_2$, 20 μM bicuculline methiodide, 50 μM hexamethonium bromide, and 5 μM mecamylamine hydrochloride at a rate of 3–4 ml/min at 32.0 ± 2.0°C. Rostral or lateral IPN nuclei were visually identified using an infrared differential interference contrast video system in a BX51 microscope (Olympus, Tokyo, Japan). Electrical signals were acquired at 10–50 kHz and filtered at 2 kHz using an EPC 10 (HEKA, Lambrecht/Pfalz, Germany) amplifier. Glass pipettes (BF150-86-10, Sutter Instrument, Novato, CA) with resistances of 3–4 MΩ were crafted using a P97 horizontal pipette puller (Sutter Instrument) and filled with internal solution containing (in mM) 130 K-Gluconate, 10 KCl, 5 MgCl$_2$, 5 MgATP, 0.2 NaGTP, 0.5 EGTA, 5 HEPES; pH 7.4 adjusted with KOH. Whole-cell patch-clamp recordings were performed in voltage-clamp mode at a holding potential of –60 mV, and access resistance was constantly monitored via a –10 mV voltage step at the end of each sweep. Recordings with access resistances exceeding 20 MΩ or with changes in access resistance or holding current by more than 20% were discarded. To evoke glutamatergic excitatory postsynaptic currents (EPSCs) in rostral IPN neurons, voltage pulses (0.5–3.5 V, 0.2 ms duration) were applied with an ISO-Flex stimulus isolator (AMPI, Jerusalem, Israel) to a concentric bipolar stimulating electrode (CBBPC75, FHC, Bowdoin, ME) located inside the IPN, ~200–300 μm distal to the recorded neuron. For optogenetic stimulation in Tac1-ChR2 mice, blue light (λ = 465 nm) was emitted directly onto the lateral IPN through a 5 mm long mono fiber-optic cannula (fiber diameter 200 μm, total diameter 230 μm, Doric lenses, Quebec, Canada) connected to a PlexBright LED (Plexon, Dallas, TX), with an optical patch cable (fiber diameter 200 μm, total diameter 230 μm, 0.48 NA). The LED was triggered via 200–290 mA current pulse (2 ms duration) from a LED Driver (LD-1, Plexon), which was controlled directly via the HEKA EPC10 amplifier. The sweep interval of all stimulation protocols (electrical and optogenetic) was 10 s. For the application of SNX-482 (1 μM, hellobio, Bristol, UK) and ω-Conotoxin GVIA (1 μM, Alomone Labs, Jerusalem, Israel), 0.1% bovine serum albumin was added to the ACSF. For variance-mean analysis, ACSF with four different Ca$^{2+}$ concentrations (1.5–8 mM) was applied to measure variance and mean EPSC amplitude at different release probabilities. Each Ca$^{2+}$ concentration was washed in for 5–10 min. Once EPSC amplitudes stabilized, 15–20 consecutive EPSC responses were used for the calculation of mean EPSC amplitude and variance, followed by the wash-in of the next Ca$^{2+}$ concentration. The order of application was 2.5 mM Ca$^{2+}$, followed by 1.5 mM Ca$^{2+}$, followed by 6–8 mM Ca$^{2+}$, followed by 4–5 mM Ca$^{2+}$. Baseline noise was much smaller than synaptic noise. The values for release probability and quantal size were calculated according to the equations Var = Iq – I$^2$/N and I = Nqp, with I being the average EPSC amplitude, N the number of release sites, and q the quantal size. To measure the paired-pulse ratio (PPR) of two consecutively evoked EPSCs at 20 Hz, the PPRs of 20–30 EPSC pairs evoked at 10 s intervals were averaged. To study the effect of GBR activation on EPSC amplitude, R(+)-Baclofen hydrochloride (1 μM) was bath applied for 3 min and washed for 10 min. For PPR and variance-mean experiments, data from wild-type C57BL/6J and BALB/c mice were found to not differ significantly and thus were pooled. To measure action potential properties in ventral MHb neurons, a 50 pA current was injected for 100 ms, which reliably evoked a train of action potentials. Action potential peak times were measured as the time from the threshold potential to the peak. Action potential full-width at half maximum was measured as the time between rising and the falling phase of the action potential at half of the maximal action potential amplitude. Only the first action potential in the train was analyzed.

## Cell lines

Human Embryonic Kidney 293T (HEK293T) were directly obtained from ATCC (https://web.expasy.org/cellosaurus/CVCL_0063)and maintained in DMEM supplemented with GlutaMAX (Invitrogen) and 10% FCS in a humidified atmosphere (5% CO$_2$) at 37°C. HEK293 cells (RRID:CVCL_0045), stably expressing the human Cav2.3 (R-type) channel α$_{1E-3}$ splice variant (also called α1E-c; GenBank L29385), the Ca$^{2+}$ channel auxiliary subunits human α$_{2b}$δ−1 (M76559) and human β$_{3a}$ (NM_000725), as well as the human potassium channel KCNJ4 (Kir2.3; U07364), were a kind gift from the

laboratory of David J. Adams (University of Wollongong, Australia). Cells were maintained in culture as described before (*Dai et al., 2008*). All cell lines were authenticated using Short Tandem Repeat (STR) analysis by Microsynth (Switzerland) and tested negative for mycoplasma contamination.

## Cell culture transfection and electrophysiology

For experiments, the Cav2.3-expressing HEK293 cells were transiently transfected with plasmids for the expression of Myc-tagged KCTD8 (AY615967) or KCTD12b (AL831725) using Lipofectamine 2000 (Invitrogen, Carlsbad, CA) as described earlier (*Schwenk et al., 2010*). Plasmids for the expression of eGFP (Clontech, Kyoto, Japan) were co-transfected in order to identify cells expressing the plasmids. A total quantity of 0.75 µg of DNA was transfected in each well, and empty pCI plasmid was used to complete to this amount of total DNA. The control group contained only eGFP and empty vector. Two to four days after transfection, cells were briefly subjected to Versene (Thermo-Fisher, Waltham, MA) treatment for its dissociation and plated on glass coverslips and, at least 2 hr later, electrophysiology experiments were performed.

Whole-cell recordings were performed using fire-polished borosilicate patch pipettes (2–4 MΩ of tip resistance), filled with a cesium-based intracellular solution containing (mM): 110 $CsMeSO_3$, 15 CsCl, 10 EGTA, 10 HEPES, 10 Tri–phosphocreatine, 0.1 NaGTP, 4 MgATP, pH 7.3 adjusted with CsOH. The extracellular solution contained (mM): 110 NaCl, 30 TEA-Cl, 10 HEPES, 10 D-glucose, 5 CsCl, 1 $MgCl_2$ and 5 mM $BaCl_2$, pH 7.4 adjusted with 40% TEA-OH. For patch-clamp experiments, isolated cells with a clear eGFP fluorescent signal were selected. The culture was maintained under continuous perfusion (~1 ml/min) at room temperature. A MultiClamp 700B (Molecular Devices, San Jose, CA) amplifier connected to a 1440A Digidata was used to acquire signals at 10–50 kHz, filtered at 2–8 kHz. After whole-cell formation, series resistance (<10 MΩ) was compensated by 85%. Linear leak currents and capacity transients were subtracted on-line using a −P/4 protocol. Current-voltage relationships were recorded from a holding potential of −80 mV using 25 ms depolarizations from −70 to 60 mV in 10 mV increments. Peak $I_{Ba}$ was measured at each step and normalized to the capacitance of the cell to obtain the current densities, which were plotted as a function of the voltage step. I−V curves were fit with a standard Boltzmann equation as follows: $I_{Ba} = G_{max}(V − V_{rev})/(1 + exp((V_{0.5act} − V)/k))$, where $I_{Ba}$ is the current measured at each test potential $V$, $G_{max}$ is the maximal conductance, $V_{rev}$ is the reversal potential, $V_{0.5act}$ is the half-maximal voltage of activation, and $k$ is the slope factor. To study voltage dependency of activation, a series of 25 ms voltage steps from −80 to 60 mV (with 10 mV increments) were applied, followed by a 10 ms step to 0 mV, where peaks of the evoked current tails were measured, normalized to the maximal peak, and plotted against the corresponding voltage. To the study-steady state inactivation, a series of 2 s long voltage steps from −120 to 10 mV (10 mV increments) was applied, followed by a test step to 10 mV (150 ms) where peaks of elicited currents were measured, normalized to the maximal response, and plotted against the corresponding voltage. Both activation and inactivation curves were fitted using a Boltzmann sigmoidal as follows: $I = I_2 + (I_1 − I_2) / (1 + exp((V_{0.5\ act/inact} − V_t)/k))$, where $V_{0.5\ act/inact}$ is the half-maximal activation or inactivation and $k$ is the slope factor that was positive for activation and negative for inactivation curves.

## Co-immunoprecipitation

Plasmids encoding N-terminally Flag-tagged KCTDs were described earlier (*Seddik et al., 2012*). Plasmids expressing C-terminally Flag-tagged Cav2.3 were obtained from GenScript. Cultured human embryonic kidney 293T (HEK293T) cells were transfected using 2 µg/µl polyethylenimine (Sigma-Aldrich) with indicated plasmids for the expression of either Flag-KCTDs alone or in combination with Flag-Cav2.3. Cells were washed 48 hr after transfection in ice-cold PBS and lysed in NETN buffer (100 mM NaCl, 1 mM EDTA, 0.5% Nonidet P-40, 20 mM Tris–HCl, pH 7.4, supplemented with cOmplete EDTA-free protease inhibitor mixture [Roche, Basel, Switzerland]), followed by rotation for 10 min at 4℃. Cell lysates were then cleared by centrifugation at 10,000 × g (4℃, 10 min) and directly used for immunoblot analysis (Input) or for co-immunoprecipitation assay, in which they were precleared for 1 hr using 30 µl (dry volume) of a 1:1 mixture of protein-A and protein-G-agarose beads (GE Healthcare, Chicago, IL). Thereafter, lysates were incubated by rotating for 16 hr at 4℃ in the presence of 2.5 µl of 0.3 µg/µl anti-Cav2.3 (CACNA1E) antibody (ACC-006, Alomone Labs, Jerusalem, Israel). The next day, 10 µl (dry volume) of a 1:1 mixture of protein-A and protein-

G-agarose beads were added and incubated by rotating for 40 min at 4°C. Afterward, they were washed with 5 × 1 ml of NETN buffer and pulled down proteins were eluted with 25 µl of 4× sample loading buffer containing 200 mM DTT. Proteins were resolved using standard one-dimensional SDS–PAGE on 10% polyacrylamide gels (for 45 min at 70 mV, followed by an additional 1.5 hr at 120 mV). For immunoblotting analysis, proteins were transferred to 0.45 µM polivinylidene fluoride membranes (Milipore, Burlington, MA) for 2 hr at 200 mA and probed with the primary antibodies rabbit anti-Flag (F7425, Sigma) and anti-Cav2.3 (ACC-006, Alomone Labs) in combination with peroxidase-coupled secondary donkey anti-rabbit antibodies (NA934, GE Healthcare, 1:10,000).

For co-IP experiments using membrane extracts, we transiently transfected HEK293T cells with the pore forming α-subunit (rat Cav2.3) together with auxiliary β3 and α2δ1 subunits in order to emulate the experimental conditions present the stable cell line (*Dai et al., 2008*). These cells were then subjected to hypotonic lysis to separate cytosolic proteins from membrane proteins (total membrane). A fraction of the total membrane pool was then subjected to a plasma membrane purification protocol using a commercially available plasma membrane extraction kit (ab 65400, Abcam, Cambridge, UK). The plasma membrane fraction obtained after the purification as well as the total membrane fraction were subsequently solubilized in detergent-containing buffer (0.5% Nonidet P-40) and loaded on 10% SDS–PAGE gels. The presence of the KCTD proteins in the corresponding tissues was determined by western blotting using ECL detection of HRP-coupled antibodies against Flag-tagged KCTD8 (rabbit anti-KCTD8) and Flag-tagged KCTD12b (mouse anti-Flag). For co-immunoprecipitation assay, lysates were incubated by rotating for 16 hr at 4°C in the presence of 2.5 µl of 0.3 µg/µl anti-Cav2.3 (CACNA1E) antibody (ACC-006, Alomone Labs, Jerusalem, Israel), washed, and eluted as described above with subsequent immunoblotting analysis performed as described above.

## Lentiviral injections

Lentiviral vectors (LV–hSyn–EGFP, LV–hSyn–3xFlag/mKCTD8[NM_175519.5]/P2A/EGFP, and LV–hSyn–3xFlag/mKCTD12b[NM_175429.4]/P2A/EGFP) were generated by the life science facility at IST Austria and by VectorBuilder GmbH (Neu-Isenburg, Germany). Bilateral viral injections were performed in male C57Bl/6J or KCTD12b KO mice at 2 months of age. Seven hundred nanoliter of virus was injected at a rate of 50 nl/min to the following coordinates: anterior/posterior: –1.42 mm from Bregma, lateral: ±0.22 mm from the midline, dorsal/ventral: –2.5 mm from the surface of the brain. Mice were allowed to recover for 10–14 days before carrying out electrophysiology experiments.

## Data analysis and statistics

All statistical tests and graph preparations were done using Prism (GraphPad, San Diego, CA), and figure assembly was done in Photoshop (Adobe, San Jose, CA). To determine whether to use parametric or non-parametric statistical tests, Shapiro-Wilk test for normality of residuals was applied. Unless otherwise noted, averaged data is presented as mean ± SEM, and p-values<0.05 were considered to indicate statistical significance. Power analysis (power = 0.8, α error = 0.05) was used to determine the appropriate sample size for SDS-FRL and variance-mean experiments using G*Power software (Univ. Kiel, Germany). Sample size in other experiments was based on those of similar experiments in previous studies. Masking was not used for any experiment. For the analysis of pre-embedding immunolabeled ultrathin sections, the presynaptic active zone was manually demarcated based on (1) rigid alignment of pre- and postsynaptic membranes and the presence of a postsynaptic density with the same length as the presynaptic active zone, as previously described (*Rubio et al., 2017*). Active zone length and silver-intensified gold particle densities and distances were measured using Reconstruct software. In SDS-FRL samples, demarcation of presynaptic active zones was performed manually based on two morphological criteria: there had to be a visible alteration in surface curvature in the P-face and/or a concentration in intramembrane particles (IMPs) of variable sizes that appeared clearly different from those of the surrounding P-face. For density measurements, incomplete active zones were analyzed whenever there was a visible transition to the postsynaptic E-face that displayed the characteristic IMP clusters of a glutamatergic postsynapse (*Tanaka et al., 2005*). For confirmation, the complete active zone area demarcated based on protoplasmic surface depression and concentrated IMP cluster were compared with the area of complete

postsynaptic E-face IMP clusters. In addition, the area of manually demarcated complete active zones was also compared with those demarcated using a mixture of active zone-marker antibodies for RIM1/2 (5 µg/ml; Synaptic Systems, Göttingen, Germany), neurexin (5 µg/ml) (*Miki et al., 2017*), and CAST (3 µg/ml) (*Hagiwara et al., 2018*). To analyze clustering of molecules (Cav2.3, GABA$_{B1}$, KCTD8, and KCTD12b), we performed 100 Monte Carlo simulations using GPDQ software (*Luján et al., 2018*). For each simulation, the same number of particles was redistributed randomly, with each pixel having the same probability of becoming the center of a particle, on the demarcated area of interest under the condition that two particles could not be closer to each other than 10 nm. We then compared nearest neighbor distances (NNDs) between real particles with the NNDs of simulated particles using Kolmogorov–Smirnov test.

## Acknowledgements

We are grateful to Akari Hagiwara and Toshihisa Ohtsuka for CAST antibody, and Masahiko Watanabe for neurexin antibody. We thank David Adams for kindly providing the stable Cav2.3 cell line. Cav2.3 KO mice were kindly provided by Tsutomu Tanabe. This project has received funding from the European Research Council (ERC) and European Commission (EC), under the European Union's Horizon 2020 research and innovation programme (ERC grant agreement no. 694539 to Ryuichi Shigemoto, no. 692692 to Peter Jonas, and the Marie Skłodowska-Curie grant agreement no. 665385 to Cihan Önal), the Swiss National Science Foundation Grant 31003A-172881 to Bernhard Bettler and Deutsche Forschungsgemeinschaft (For 2143) and BIOSS-2 to Akos Kulik.

## Additional information

### Funding

| Funder | Grant reference number | Author |
| --- | --- | --- |
| H2020 European Research Council | 694539 | Ryuichi Shigemoto |
| H2020 European Research Council | 692692 | Peter Jonas |
| Schweizerischer Nationalfonds zur Förderung der Wissenschaftlichen Forschung | 31003A-172881 | Bernhard Bettler |
| H2020 Marie Skłodowska-Curie Actions | 665385 | Cihan Önal |
| Deutsche Forschungsgemeinschaft | For2143 | Akos Kulik |
| Deutsche Forschungsgemeinschaft | BIOSS-2 | Akos Kulik |

The funders had no role in study design, data collection and interpretation, or the decision to submit the work for publication.

### Author contributions

Pradeep Bhandari, Data curation, Formal analysis, Investigation, Methodology, Writing - review and editing; David Vandael, Diego Fernández-Fernández, Thorsten Fritzius, Data curation, Formal analysis, Investigation, Writing - review and editing; David Kleindienst, Software, Formal analysis, Writing - review and editing; Cihan Önal, Data curation, Writing - review and editing; Jacqueline Montanaro, Methodology, Writing - review and editing; Martin Gassmann, Resources, Writing - review and editing; Peter Jonas, Akos Kulik, Resources, Funding acquisition, Writing - review and editing; Bernhard Bettler, Resources, Supervision, Funding acquisition, Writing - review and editing; Ryuichi Shigemoto, Conceptualization, Resources, Supervision, Funding acquisition, Writing - original draft, Project administration, Writing - review and editing; Peter Koppensteiner, Conceptualization, Data curation, Formal analysis, Investigation, Writing - original draft, Writing - review and editing

## Author ORCIDs

Diego Fernández-Fernández (iD) http://orcid.org/0000-0003-1431-3705
Thorsten Fritzius (iD) https://orcid.org/0000-0002-3597-6623
Cihan Önal (iD) http://orcid.org/0000-0002-2771-2011
Peter Jonas (iD) http://orcid.org/0000-0001-5001-4804
Ryuichi Shigemoto (iD) https://orcid.org/0000-0001-8761-9444
Peter Koppensteiner (iD) https://orcid.org/0000-0002-3509-1948

## Ethics

Animal experimentation: All experiments were performed in accordance with the license approved by the Austrian Federal Ministry of Science and Research (Animal license number: BMWFW-66.018/0012-WF/V/3b/2016) and the Austrian and EU animal laws.

## Decision letter and Author response

Decision letter https://doi.org/10.7554/eLife.68274.sa1
Author response https://doi.org/10.7554/eLife.68274.sa2

## Additional files

### Supplementary files

• Supplementary file 1. Contains Table 1 showing parameters of the Boltzmann fit shown in *Figure 8C*. $G_{max}$ is the maximal conductance density. *$G_{max}$ was significantly increased in KCTD8-transfected cells compared with Control (p=0.0340, one-way ANOVA with Tukey post hoc test); $V_{rev}$ is the reversal potential; $V_{0.5\ act}$ is the potential at which current density was half-maximal; $K_{\alpha}$ is the slope factor.

• Transparent reporting form

### Data availability

For all figures, numerical data that are represented in graphs are provided as source data excel files.

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
