## [Decision Letter]

**Acceptance summary:**

This study examines presynaptic regulation of the medial habenula to interpeduncular nucleus connection, which plays a crucial role in aversion and addiction related behaviors. The authors focus on the special features of these synapses, namely the presynaptic expression of R-type voltage gated calcium channel, Cav2.3, the potent modulation by the presynaptic GABA_B_ receptors, and their auxiliary KCTD subunits whose functions have been elusive. Through electrophysiological, biochemical and ultrastructural characterizations, the paper shows that KCTD8 and KCTD12b directly interact with Cav2.3 and regulate neurotransmitter release from ventral medial habenula terminals independently of GABA_B_ receptors, where effectively KCTD12b serves to restrain release while the recruitment of KCTD8 to the active zone potentiates release by promoting Cav2.3 currents. These findings are important as they identify novel, GABA_B_ receptor-independent role for KCTD proteins in regulating neurotransmission and provide insights into the underlying mechanisms.

**Decision letter after peer review:**

[Editors’ note: the authors submitted for reconsideration following the decision after peer review. What follows is the decision letter after the first round of review.]

Thank you for submitting your work entitled "GABA_B_ receptor auxiliary subunits modulate Cav2.3-mediated release from medial habenula terminals" for consideration by *eLife*. Your article has been reviewed by 3 peer reviewers, one of whom is a member of our Board of Reviewing Editors, and the evaluation has been overseen by a Senior Editor. The reviewers have opted to remain anonymous.

Our decision has been reached after consultation between the reviewers. Based on the discussions and the individual reviews below, we regret to inform you that your work will not be considered further for publication in *eLife*, at least in its present form.

All three reviewers recognized the data are of very high quality and the findings to be potentially interesting. However, concerns were raised with respect to uncertainties about the underlying mechanisms of GABA_B_ receptor-mediated presynaptic modulation and the roles of KCTDs, such that the study remained premature at this stage.

*Reviewer #1:*

This study examines presynaptic modulation of the medial habenula (MHb) to interpeduncular nucleus (IPN) connection, which plays a crucial role in aversion and addiction related behaviors. The authors focus on the special features of these synapses, namely the expression of R-type VGCC, Cav2.3 in presynaptic active zones and the robust presynaptic GABA_B_ receptor (GBR)-mediated potentiation. In particular, they seek to determine the mechanism of GBR-dependent potentiation of release by considering the actions of the KCTD subfamily members that serve as auxiliary subunits of GBR.

The study reveals in presynaptic terminals of ventral MHb to rostral IPN connection in KCTD12b KO mice, a gain of function activity of KCTD8 in potentiating neurotransmitter release, likely via interacting with Cav2.3 to modulate its channel activity. It remains unclear what the normal functions of KCTD8 and KCT12b might be. For example, the lateral IPN has no detectable KCTD12b but expresses high levels of KCTD8 (Figure 2), yet its response to baclofen is opposite to the response detected in the rostral IPN in KCTD12b KO mice. Given that KO mice have been analyzed in the adult stage, whether the developmental compensatory effects could have contributed to the observed effects, requires attention also.

Altogether, the study provides an informative characterization of the ultrastructural localization of KCTD proteins, GBR and Cav2.3 at the synapse, and neurotransmitter release properties of synapses lacking KCTD8 and KCTD12b either singly or in combination. The paper is written clearly and the experiments have been carefully executed. However, further experiments and analysis are required to compellingly support the main conclusions and establish further insights into the mechanism by which KCTD8 and KCTD12 regulate basal release and clarify their contributions to GBR-dependent release.

Specific points

If there is a bi-directional regulation of neurotransmitter release by KCTDs in which KCTD8 potentiates whereas KCTD12b depresses, then one should test the effects of overexpression of KCTD8 and KCTD12b in WT animals to evaluate the competitive nature of the actions of KCTD8 and KCTD12b.

In order to address whether there may be developmental contribution of KCTD8 and KCTD12b, one should test whether acute knock-down produces the same effects and/or demonstrate the sufficiency by rescuing the mutant phenotype with exogenous overexpression in the adult.

p. 4, middle paragraph: "KCTD subunits co-precipitate with release machinery proteins of the presynaptic active zone…" Please specify which active zone proteins?

In KCTD8 KO neurons, is there a compensatory recruitment of KCTD12b to the active zone, that is, the opposite of what takes place in KCTD12b KO neurons?

Figure 2 – it would be useful to test for cross-labelling to evaluate any compensatory expression for the loss of particular KCDT subtype with another KCDT subtype at the level of light microscopy.

Figure 6A. Regarding the representative plot for the double KO mice, it is worrisome that there is a drastic increase in the baseline prior to baclofen addition, such that it is difficult to interpret the rest of the data points. Does baclofen still potently increase EPSC amplitudes in KCTD8, KCTD12b KO or KCTD8/12b DKO mice if recordings are carried out in 4 mM extracellular Ca instead of 2.5 mM?

Figure 7. To determine the nature of the increase in the active zone expression of KCTD8 in KCTD12b KO terminals, one should perform (1) the distance-dependent analysis of immunogold labelling for KCTD8 in KCTD12b KO preparation as shown for Figure 3, especially with respect to whether the profile of KCTD8 changes to what is shown for KCTD12b in rostral IPN, and (2) analysis at the level of light microscopy to assess whether there is an increase in the level of expression or KCTD8 in KCTD12b KO rostral IPN (i.e. the same comment as for Figure 2 above).

Figure 8. The relative levels of surface expression of KCTD8 and KCTD12b in HEK cells needs to be quantified in order to fully interpret the preferential increase in Cav2.3-mediated currents in KCTD8-transfected cells. The direct interaction between KCTD8 and KCTD12b with Cav2.3 should be confirmed in the brain.

*Reviewer #2:*

This manuscript aims to examine the mechanism of the unusual potentiating effect of baclofen acting on presynaptic GABA_B_ receptors in the MHb-IPN pathway. I was expecting an examination of underlying second messenger pathways, but this was not the route taken, as the authors investigating whether there is a role for the KCTD proteins which interact with GABA_B_ receptors.

My impression of this manuscript is that by trying to understand the potentiating effect of baclofen on the synapses in question, the authors have uncovered a very interesting effect of KCTD8 potentiating CaV2.3, and an interplay between KCTD8 and KCTD12b, shown with several techniques including very nice EM images and analysis, which however is completely separate from the effect of baclofen acting on GABA_B_ receptors.

In trying nevertheless to combine the two stories, the authors have overcomplicated the message of their study, in my view. I would prefer to see the results separated, with the KCTD study stripped out and potentially published here, as it is novel and well performed. Otherwise it seems to me the manuscript is over-complex and the very interesting message about the KCTDs is difficult to extract.

1. Line 99 SNX482 is not "selective". The authors are using it at 1 µM. It may be relatively selective between calcium channels, but it also blocks certain K channels in the same concentration range, in fact 500 nM was used in slices to block A-type K currents (Kimm and Bean, 2014). It also affects other Ca channels and Na channels at concentrations below 1µM (Newcomb et al., 1998). How sure are the authors this blockade of other channels is not affecting their study ?

2. Line 104. You cannot say the two MHb-IPN pathways have different sensitivities to SNX482, as no dose response curve has been performed, and the difference between the pathways at the single dose studied may indicate the involvement of other Ca channels.

3. Similarly, in the Discussion, line 310 "We confirmed that both dorsal MHb-IPN and ventral MHb-IPN pathways require Cav2.3 for transmitter release", is not completely correct, as how is release occurring for the ~50% SNX482-insensitive lateral epscs (Figure 1) ?

4. Figure 8A needs correct labelling of the panels, with MW markers. I assume the top panel is KCTDs, but this is unclear as both CaV2.3 and KCTDs have Flag tags, which is an unusual way of performing the experiment. The entire blot should be shown for the top panel, as blotting with Flag would show both KCTDs and CaV2.3.

5. If there is co-ip between CaV2.3 and KCTD8/12b, should there be actual co-localization observed in the EM images? Can this be measured?.

6. In Figure 8c, how selective is the augmentation with KCTD8 and CaV2.3, ie does it also occur for any other Ca channels? It is possible that a low level of other presynaptic Ca channels might be also augmented by KCTD8 in these terminals, increasing Ca^2+^ entry.

7. If this large augmentation of CaV2.3 currents by KCTD8 plays a physiological role should the epsc's in the KCTD8 KO mice be reduced? I cannot see this basic result, although I note the release probability is indeed reduced.

8. Following on from point 6, in Discussion line 329, has the 1997 study by Ludwig et al. showing no α1B, A, C mRNA in the habenula been replicated anywhere? The technique used in this early study is not very sensitive, are there other more recent studies using single cell RNAseq for example?

9. Discussion line 330 "Although this exclusivity has been described anatomically (Parajuli et al., 2012)" is not correct as it gives the impression other Ca channels were examined in this study, whereas this is not the case.

10. Discussion line 331: SNX482 is here described as specific, which is incorrect.

References

Kimm, T., and Bean, B.P. (2014). Inhibition of A-type potassium current by the peptide toxin SNX-482. J Neurosci 34, 9182-9189.

Newcomb, R., Szoke, B., Palma, A., Wang, G., Chen, X.H., Hopkins, W., Cong, R., Miller, J., Urge, L., Tarczy-Hornoch, K., et al. (1998). Selective peptide antagonist of the class E calcium channel from the venom of the tarantula Hysterocrates gigas. Biochemistry 37, 15353-15362.

*Reviewer #3:*

The manuscript attempts to elucidate the mechanisms underlying the potentiation of synaptic transmission between the ventral medial habenula (MH) and the rostral interpeduncular nucleus mediated by presynaptic GABA-B receptors. The topic is interesting especially because it is very unusual that GABA-B receptors gate an enhancement of neurotransmission, since usually when they are activated, they depress the release of neurotransmitter.

The manuscript contains some high quality data, for example, freeze-fracture replica immunolabelling of presynaptic proteins in habenular terminals. However, I also found some technical problems, (e.g., see comment #1 below). Overall, the manuscript is potentially interesting and covers an original topic. However, the data currently submitted fail to answer the two major questions, namely the underlying mechanisms and the physiological relevance of GABA-B receptor-mediated modulation. A lot of efforts are devoted in characterizing key protein expression in various K^+^ channel tetramerization domain-containing (KCTD) proteins. However, I doubt that this strategy paid off and really shed light on the mechanisms underlying GABA-B receptor mediated modulation of the synapses studied.

1. Data on the action of SNX-482 on evoked EPSCs need refinement. The presynaptic fibers of the fasciculus retroflexus were electrically stimulated and recordings were made in the rostral IPN. In contrast, stimulation of dorsal fibers were obtained via optogenetic stimulation and responses were recorded in the lateral IPN. The effect of SNX-482 was stronger at the former versus the latter synapses. However, this result is at odds with similar Cav2.3 densities in the active zones of dorsal and ventral MHb terminals. The Authors discussed this issue, and I acknowledged this mis-match in their data. Nevertheless, the problem remains: a quantitative comparison between drug modulation of synaptic responses evoked by electrical versus optogenetic stimulation is a poor choice. Amongst various problems, several recent papers report the non-physiological high release probability often caused by optogenetic stimulation of presynaptic fibers that contrasts with the electrical stimulation method. This discrepancy would affect the results obtained with SNX-482. This set of experiments should be repeated by using more uniform experimental conditions in the two synaptic pathways. Furthermore, a serious comparison of the efficacy of SNX-482 should rely on testing at least three different drug concentrations for each synapse. Furthermore, normalization assuring that approximatively an equal number of presynaptic fibers are stimulated at each synaptic connection should be employed. What ever methods of presynaptic stimulation will be used, optical or electrical, test of selective stimulation of fibers should be also performed.

2. Baclofen still presynaptically potentiated EPSCs in KCTD KO mice. However, loss of KCTD12b impaired GABABR action probably via increasing release probability and occlusion. In contrast, lack of KCTD8 reduces release probability. But what about the effects of GABABR modulation? Is baclofen more powerful at MH synapses in KCTD8 KO?

3. The manuscript lacks to experimentally demonstrate that GABA-B receptor potentiation of ventral MH and the rostral interpeduncular nucleus synapses is physiologically relevant. This should be addressed by using physiologically salient repetitive stimulation in control and in the presence of GABA-B R antagonists. It would be also interesting to test the GABA-B R ligands on evoked EPSPs followed by IPSPs to see the comprehensive pharmacological effects on both excitation and inhibition. At present none of these important points are addressed and this reduces the impact of the data currently submitted.

[Editors’ note: further revisions were suggested prior to acceptance, as described below.]

Thank you for submitting your article "GABA_B_ receptor auxiliary subunits modulate Cav2.3-mediated release from medial habenula terminals" for consideration by *eLife*. Your article has been reviewed by 3 peer reviewers, one of whom is a member of our Board of Reviewing Editors, and the evaluation has been overseen by John Huguenard as the Senior Editor. The following individual involved in review of your submission has agreed to reveal their identity: Marco Capogna (Reviewer #2).

The revised manuscript has been significantly improved by refocusing the study on the role for KCTDs on neurotransmitter release. Several additional experiments have been performed, including the comparisons of Cav2.3-dependence of synaptic transmission between the two distinct medial habenula to IPN inputs by electrical stimulation, which has been made possible by the establishment of thick-slice preparation. New data lend further support to the conclusion that KCTD8 and KCTD12b directly interact with Cav2.3 and regulate release from ventral MHb terminals independently of GABA_B_ receptors, where effectively KCTD12b serves to restrain release while the recruitment of KCTD8 to the active zone potentiates release by promoting Cav2.3 currents. Prior to accepting the manuscript for publication, a number of points require attention.

Figure 6F-G: the EPSC waveforms of some rescued currents appear to be quite different than one would expect. Specifically, the LV-hSyn-KCTD8 current is quite prolonged, and may have two components to the rising phase. Additionally, the LV-hSyn-KCTD12b current appears to have a delayed onset. Are these examples representative of the population? If so, are these effects a result of over-expression, or more an indicator of the distinct roles of these two KCTD proteins?

In the introduction (Line 47 or Line 53-54), it would be clearer to readers who may not necessarily be familiar with KCTDs if the authors could briefly state the nature of the molecular difference especially between KCTD12 and KCTD12b.

Line 302-304. "The decrease in release probability by KCTD12b in the presence but not in the absence of KTCD8…" is confusing. Do the authors mean that the reason why the decreased PPR observed in KCTD12bKO could be recovered to control levels when KCTD8 is additionally absent (as observed in double KO mice), could be because the loss of KCTD12b in itself alters the localization of KCTD8 or Cav2.3? Perhaps this sentence should be rephrased.

---

## [Author Response]

[Editors’ note: the authors resubmitted a revised version of the paper for consideration. What follows is the authors’ response to the first round of review.]

Reviewer #1:This study examines presynaptic modulation of the medial habenula (MHb) to interpeduncular nucleus (IPN) connection, which plays a crucial role in aversion and addiction related behaviors. The authors focus on the special features of these synapses, namely the expression of R-type VGCC, Cav2.3 in presynaptic active zones and the robust presynaptic GABA_B_ receptor (GBR)-mediated potentiation. In particular, they seek to determine the mechanism of GBR-dependent potentiation of release by considering the actions of the KCTD subfamily members that serve as auxiliary subunits of GBR.The study reveals in presynaptic terminals of ventral MHb to rostral IPN connection in KCTD12b KO mice, a gain of function activity of KCTD8 in potentiating neurotransmitter release, likely via interacting with Cav2.3 to modulate its channel activity. It remains unclear what the normal functions of KCTD8 and KCT12b might be. For example, the lateral IPN has no detectable KCTD12b but expresses high levels of KCTD8 (Figure 2), yet its response to baclofen is opposite to the response detected in the rostral IPN in KCTD12b KO mice. Given that KO mice have been analyzed in the adult stage, whether the developmental compensatory effects could have contributed to the observed effects, requires attention also.Altogether, the study provides an informative characterization of the ultrastructural localization of KCTD proteins, GBR and Cav2.3 at the synapse, and neurotransmitter release properties of synapses lacking KCTD8 and KCTD12b either singly or in combination. The paper is written clearly and the experiments have been carefully executed. However, further experiments and analysis are required to compellingly support the main conclusions and establish further insights into the mechanism by which KCTD8 and KCTD12 regulate basal release and clarify their contributions to GBR-dependent release.

We have re-structured our manuscript now focusing on the role of the KCTDs on Cav2.3-mediated basal release, and added further experiments and analysis to support our initial claims as follows.

Specific pointsIf there is a bi-directional regulation of neurotransmitter release by KCTDs in which KCTD8 potentiates whereas KCTD12b depresses, then one should test the effects of overexpression of KCTD8 and KCTD12b in WT animals to evaluate the competitive nature of the actions of KCTD8 and KCTD12b.

We thank the reviewer for raising this important point. We do not think that KCTD12b is actively depressing release since it did not affect Cav2.3-mediated currents in vitro. We rather think that KCTD8 is enhancing release by increasing currents through Cav2.3 and KCTD12b may just compete with KCTD8 for binding to Cav2.3 and prevent the effect of KCTD8. To test the effect of overexpression of KCTD8 on basal release from ventral MHb terminals, we injected lentivirus expressing either EGFP or KCTD8-p2A-EGFP bilaterally into the MHb of WT mice (Figure 6F). PPR values recorded in rostral IPN neurons of EGFP control-injected animals remained unchanged (LV-EGFP: 2.17 ± 0.18, n = 17, 5 mice), whereas KCTD8-injected animals exhibited significantly lower PPR values (LV-KCTD8: 1.41 ± 0.14, n = 13, 3 mice; t_28_ = 3.20, P = 0.0034; unpaired t-test). These results are described on page 14, starting from line 288.

In order to address whether there may be developmental contribution of KCTD8 and KCTD12b, one should test whether acute knock-down produces the same effects and/or demonstrate the sufficiency by rescuing the mutant phenotype with exogenous overexpression in the adult.

To address this important point, we used lentiviral expression of KCTD12b in MHb neurons of KCTD12b KO mice. Lentiviral expression of KCTD12b-p2A-EGFP in KCTD12b KO mice significantly increased PPR values compared with EGFP control-injected animals (Figure 6G; KCTD12b KO EGFP Control: 1.38 ± 0.09, n = 11, 3 mice; PPR KCTD12b KO rescue: 2.18 ± 0.29, n = 14, 3 mice; t_23_ = 2.98, P = 0.0068, unpaired t-test), rescuing the mutant phenotype (no significant difference in PPR compared with wild-type mice shown in Figure 6A; WT: 2.17 ± 0.18, n = 17, 5 mice; KCTD12b KO rescue: 2.18 ± 0.29, n = 14, 3 mice; t_29_ = 0.40, P = 0.6925, unpaired t-test). These results are described on page 14, starting from line 293.

p. 4, middle paragraph: "KCTD subunits co-precipitate with release machinery proteins of the presynaptic active zone…" Please specify which active zone proteins?

We apologize for the lack of detail. We included a more detailed description of the findings from the affinity purifications in the cited paper. The updated sentence now reads “Based on proteomics studies, voltage-gated Ca^2+^ channels co-precipitate GBRs and their auxiliary KCTD subunits together with release machinery proteins of the presynaptic active zone, including SNAP-25, synaptotagmins, synaptobrevin-2, Munc13-1, syntaxins, RIM1 and synapsins, among others (Müller et al., 2010).” This sentence is found in the introduction section on page 4, lines 59 – 62.

In KCTD8 KO neurons, is there a compensatory recruitment of KCTD12b to the active zone, that is, the opposite of what takes place in KCTD12b KO neurons?

This is an important question. It is unclear whether a compensatory increase in KCTD12b occurs in KCTD8 KO mice. Our initial results from WT pre-embedding EM indicate that KCTD12b displays peak densities in the active zone whereas KCTD8 shows peak densities in the perisynaptic region. Therefore, although it may be possible to see a compensatory increase in KCTD12b in the active zone of KCTD8 KO mice, due to the weaker expression of KCTD8 in the WT active zone, such an increase would be expected to be smaller than the compensatory increase of KCTD8 seen in KCTD12b KO animals. Furthermore, the functional consequence of a compensatory increase in KCTD12b in the active zone in KCTD8 KO mice would not be detected in functional experiments, as KCTD12b did not alter Cav2.3 function in vitro and KCTD12b has no effects on PPR in the absence of KCTD8 (no significant difference between KCTD8 KO and KCTD8/12b double KO in Figure 6A). Our results suggest that the relatively mild reduction in release probability seen in KCTD8 KO mice resulted from removal of the comparatively low levels of KCTD8 from the WT active zone, rather than from a compensatory increase in KCTD12b. We added these thoughts to the discussion on page 21, starting from line 439.

Figure 2 – it would be useful to test for cross-labelling to evaluate any compensatory expression for the loss of particular KCDT subtype with another KCDT subtype at the level of light microscopy.

We were unable to detect significant differences in fluorescence intensities for KCTD8 in KCTD12b KO animals. This might have been due to potential postsynaptic expression of KCTD8 as well as difficulty to detect nanoscale changes in protein distribution on the level of light microscopy. Instead, to confirm the compensatory increase of KCTD8 in ventral MHb terminals in the absence of KCTD12b, we performed pre-embedding EM immunolabeling for KCTD8 in presynaptic terminals in rostral and lateral IPN of KCTD12b KO mice (Figure 7C). The distribution pattern of KCTD8 in the lateral IPN showed peak densities in the peri-synaptic region, as seen in WT (Figure 3C). In contrast in the rostral IPN, the density of KCTD8 particles in the active zone, but not in the peri- and extrasynaptic area, was significantly higher compared with lateral IPN synapses (main effect of presynaptic distribution: F_4, 775_ = 31.65, P < 0.0001; rostral AZ vs. lateral AZ: P < 0.0001; two-way ANOVA with Bonferroni post hoc test). This observation confirmed that KCTD8 compensates for the loss of KCTD12b selectively in the active zone of ventral MHb terminals. This was added to the Results section on page 16, starting from line 329.

Figure 6A. Regarding the representative plot for the double KO mice, it is worrisome that there is a drastic increase in the baseline prior to baclofen addition, such that it is difficult to interpret the rest of the data points.

We apologize for unfortunate choice in the example time course for the double KO. Upon close inspection of the example time course in question, we realized that the baseline prior to baclofen was actually stable with the exception of the first minute of the recording (min 2 – 5 showed no tendency to ramp up). We therefore excluded the first 5 sweeps of this baseline recording. Using linear regression, we confirmed that the slope of the regression line over the remaining baseline points was not significantly deviating from 0 (P = 0.1318). We updated the corresponding relative values which were included in the graph, now moved to Figure 6—figure supplement 1B.

Does baclofen still potently increase EPSC amplitudes in KCTD8, KCTD12b KO or KCTD8/12b DKO mice if recordings are carried out in 4 mM extracellular Ca instead of 2.5 mM?

Although we did not test the effect of baclofen at higher Ca^2+^ concentrations in KCTD KO mice, we think it is a likely possibility that potentiation still occurs at increased external Ca^2+^ levels. However, since other reviewers felt the paper suffers from focusing too much on GBR-mediated potentiation instead of the role of KCTDs in basal release, we moved the baclofen experiments in KCTD KO mice from the main text to the supplement, and completely removed the analysis and discussion of alterations in potentiation time course by KCTDs and increased Ca^2+^ concentrations. We will deal with the mechanism of GBR-mediated potentiation in a separate paper.

Figure 7. To determine the nature of the increase in the active zone expression of KCTD8 in KCTD12b KO terminals, one should perform (1) the distance-dependent analysis of immunogold labelling for KCTD8 in KCTD12b KO preparation as shown for Figure 3, especially with respect to whether the profile of KCTD8 changes to what is shown for KCTD12b in rostral IPN, and (2) analysis at the level of light microscopy to assess whether there is an increase in the level of expression or KCTD8 in KCTD12b KO rostral IPN (i.e. the same comment as for Figure 2 above).

Please see our detailed answer to point 5. In short, we performed the suggested measurements of pre-embedding labeling and found a significant increase in KCTD8 densities in the AZ of rostral but not lateral terminals in KCTD12b KO mice.

Figure 8. The relative levels of surface expression of KCTD8 and KCTD12b in HEK cells needs to be quantified in order to fully interpret the preferential increase in Cav2.3-mediated currents in KCTD8-transfected cells.

To address this issue, we performed a plasma membrane purification experiment to determine the relative levels of surface expression of KCTD8 and KCTD12b in HEK cells. For this we transiently transfected HEK293T cells with the pore forming α-subunit (rat Cav2.3) together with auxiliary β3 and α2δ1 subunits in order to emulate experimental conditions that are as close as possible to the stable cell line used for the electrophysiology experiments in Figure 2F – H. These cells were then subjected to hypotonic lysis to separate cytosolic proteins from membrane proteins (a fraction that we called TM or “total membrane”). A fraction of the TM was then subjected to a plasma membrane purification protocol using a commercially available plasma membrane (PM) extraction kit from Abcam (ab 65400). We decided to work with this kit due to its extensive use in peer reviewed publications (so far it has been used in 85 papers – 10 of which were published between 2019 and 2020). This kit separates proteins of the plasma membrane from proteins located on other cellular membranes or the cytosol, using a 2-phase aqueous polymer method that takes advantage of differences in surface properties of the membranes rather than size and density. The PM fraction obtained after the purification as well as the TM fraction were subsequently solubilized in detergent-containing buffer (0.5% Nonidet P-40) and loaded on 10% SDS-PAGE gels. The presence of the KCTD proteins in the corresponding tissues was determined by Western Blotting using ECL detection of HRP-coupled antibodies against Flag-tagged KCTD8 (α-KCTD8 rabbit) and Flag-tagged KCTD12b (α-Flag mouse).

The following has been added to the Results section on page 8, starting from line 171.

“Plasma membrane preparations (see Methods) revealed that in HEK293T cells co-expressing Cav2.3 and KCTDs, both KCTD8 and KCTD12b are highly and similarly enriched in plasma membrane extracts, when compared to total membrane extracts [Figure 2—figure supplement 1; KCTD8: 10.7-fold (± 1.9), KCTD12b: 12.7-fold (± 2.6), no significant difference between KCTD8 and KCTD12b, P = 0.34, unpaired t-test]. We next performed co-IP experiments using total and plasma membrane extracts of transfected HEK293T cells and found that both KCTD8 and KCTD12b associate with Cav2.3 at the plasma membrane (Figure 2E).”

The direct interaction between KCTD8 and KCTD12b with Cav2.3 should be confirmed in the brain.

We agree that it would be ideal to show the direct binding of KCTDs to Cav2.3 in the brain. Unfortunately, due to technical limitations, we realized that we are unable to perform such an experiment for two main reasons:

– The IPN is relatively small (~1 x ~1 x ~0.7 mm) and weighs approximately 0.6 – 1 mg. Based on earlier Co-IP experiments that we performed in order to look at the binding of endogenous KCTDs to other interacting proteins, such as GBRs, we estimate that it would be necessary to pool tissue from over 100 mice. This constitutes an unreasonable number of animals to perform this experiment just once, and replicates might be required to consolidate findings.

– Alternatively, a co-IP could be performed from full brain lysates but this approach comes with the drawback that Cav2.3 is expressed in many brain regions whereas KCTD8 and KCTD12b show much more restricted expression. It would be theoretically possible to perform the pull down for the KCTDs, however, this runs into the problem that endogenous Cav2.3 is notoriously difficult to detect in lysates and even more challenging to detect in pull downs as co-IPs.

Due to these challenges, we designed an alternative experiment to test the interaction of Cav2.3 with KCTDs in MHb terminals. Specifically, we tested whether antibodies bound to KCTD8 might interfere with labeling of Cav2.3 by steric hindrance in KCTD12b KO mice, where KCTD8 is strongly localized to the active zone. We hypothesized that the compensatory increase in KCTD8 in the AZ might lead to increased binding of KCTD8 to Cav2.3 and, therefore, pre-incubation with anti-KCTD8 antibody might reduce the ability of the anti-Cav2.3 antibody to bind to Cav2.3, resulting in lower labeling densities. In rostral IPN synapses, we found that pre-incubation significantly reduced Cav2.3 labeling densities in the AZ but not the peri- or extrasynaptic regions (Figure 7D; main effect of synaptic distribution: F_4, 880_ = 31.42, P < 0.0001; control AZ vs. pre-incubation AZ: P = 0.0006; two-way ANOVA with Bonferroni post hoc test). In the lateral IPN, pre-incubation slightly but significantly reduced Cav2.3 labeling density only in the peri-synaptic region (Figure 7E; main effect of synaptic distribution: F_4, 890_ = 39.01; control <50 nm vs. pre-incubation <50 nm: P = 0.0320). The peri-synaptic region is the location exhibiting peak expression for KCTD8 in dorsal MHb terminals. The significant reduction in Cav2.3 labeling density at this position therefore validates our steric hindrance approach.

These results indicate a close association between KCTD8 and Cav2.3 in situ, if not their direct binding. This point was added to the Results section on page 16, starting from line 340, and discussed on page 20, staring from line 417.

Reviewer #2:This manuscript aims to examine the mechanism of the unusual potentiating effect of baclofen acting on presynaptic GABA_B_ receptors in the MHb-IPN pathway. I was expecting an examination of underlying second messenger pathways, but this was not the route taken, as the authors investigating whether there is a role for the KCTD proteins which interact with GABA_B_ receptors.My impression of this manuscript is that by trying to understand the potentiating effect of baclofen on the synapses in question, the authors have uncovered a very interesting effect of KCTD8 potentiating CaV2.3, and an interplay between KCTD8 and KCTD12b, shown with several techniques including very nice EM images and analysis, which however is completely separate from the effect of baclofen acting on GABA_B_ receptors.In trying nevertheless to combine the two stories, the authors have overcomplicated the message of their study, in my view. I would prefer to see the results separated, with the KCTD study stripped out and potentially published here, as it is novel and well performed. Otherwise it seems to me the manuscript is over-complex and the very interesting message about the KCTDs is difficult to extract.

We appreciate very much this advice and fully agree with the reviewer’s suggestion. Accordingly, we have re-structured our manuscript mostly focusing on the role of KCTDs on the Cav2.3-mediated basal neurotransmission in the MHb-IPN pathway. We will deal the mechanism of GABA_B_ receptor-mediated potentiation in a separate paper.

1. Line 99 SNX482 is not "selective". The authors are using it at 1 µM. It may be relatively selective between calcium channels, but it also blocks certain K channels in the same concentration range, in fact 500 nM was used in slices to block A-type K currents (Kimm and Bean, 2014). It also affects other Ca channels and Na channels at concentrations below 1µM (Newcomb et al., 1998). How sure are the authors this blockade of other channels is not affecting their study ?

Thank you for highlighting this important point. We agree that the concentration that we used in our acute slice preparation is higher than the concentrations used in the papers cited by the reviewer, which used either cell culture (Kimm and Bean, 2014) or oocyte preparations (Newcomb et al. 1998). Although slices were cut in the paper by Kimm and Bean 2014, they were only used to extract substantia nigra cells to prepare primary neuronal cultures, not for pharmacological experiments. An important consideration, in this respect, is the fact that SNX-482 is a peptide which may easily adhere to surfaces such as biological membranes, thereby potentially reducing its effective concentration considerably, particularly in acute brain slices. Use of SNX-482 in cell culture or oocytes naturally does not run into such issues as the pharmacological targets are readily exposed to the extracellular solution containing the drug. To reduce loss of SNX-482 as much as possible, we added 0.1 % BSA to the ACSF but it is still likely that the effective SNX-482 concentration inside the acute slice at the site of active terminals was considerably lower than 1 µM.

In addition, a block of A-type K^+^ channels would be expected to increase membrane excitability and/or action potential half-width, both of which would result in increased neurotransmitter release. However, we did not observe increases in EPSC amplitudes in any of the SNX experiments, suggesting that the observed decrease in EPSC amplitudes by the application of SNX-482 was not mediated by an inhibition of A-type K^+^ channels.

To test whether the decrease in EPSC amplitude caused by the application of SNX-482 resulted from a block of voltage-gated Na^+^ or K^+^ channels, we measured action potential properties in ventral MHb neurons in ACSF containing 0.1% BSA and in ACSF containing 0.1% BSA + 1 µM SNX-482. We defined the action potential peak time as the time between the threshold potential and the action potential peak. Action potential full-width at half maximum was measured as the time between the rising and falling phase of the action potential at half of the maximal action potential amplitude. SNX-482 did not significantly affect action potential peak times or full-widths at half maximum in ventral MHb neurons (Figure 1—figure supplement 1A; peak times: control: 0.86 ± 0.03 ms, n = 3; SNX-482: 0.93 ± 0.05 ms, n = 4; t_5_ = 1.083, P = 0.3282; full-widths: control 1.08 ± 0.07 ms, n = 3; SNX-482: 1.06 ± 0.03 ms, n = 4, t_5_ = 0.3852, P = 0.7159; unpaired t-test), suggesting that the reduction of neurotransmitter release by SNX-482 was not caused by a blockade of voltage-gated Na^+^ or K^+^ channels. This part was added to the Results section on page 6, starting from line 107.

2. Line 104. You cannot say the two MHb-IPN pathways have different sensitivities to SNX482, as no dose response curve has been performed, and the difference between the pathways at the single dose studied may indicate the involvement of other Ca channels.

We apologize for the poor wording. We removed this sentence.

3. Similarly, in the Discussion, line 310 "We confirmed that both dorsal MHb-IPN and ventral MHb-IPN pathways require Cav2.3 for transmitter release", is not completely correct, as how is release occurring for the ~50% SNX482-insensitive lateral epscs (Figure 1) ?

We changed this sentence to: “We confirmed that Cav2.3 is involved in mediating transmitter release at both MHb-IPN pathways, but only release at the ventral pathway exclusively relies on Cav2.3.” page 17, starting from line 364.

4. Figure 8A needs correct labelling of the panels, with MW markers. I assume the top panel is KCTDs, but this is unclear as both CaV2.3 and KCTDs have Flag tags, which is an unusual way of performing the experiment. The entire blot should be shown for the top panel, as blotting with Flag would show both KCTDs and CaV2.3.

We apologize for this oversight and added MW markers to the co-IP blots, now moved to Figure 2. Unfortunately, due to substantial differences in molecular weight, KCTDs and Cav2.3 cannot be run on the same gels because the high molecular weight of Cav2.3 requires a less dense gel which would not allow for adequate band separation of the significantly smaller KCTDs, while in gels with higher density that allow us to better visualize the size differences between the individual KCTDs, large molecules like Cav2.3 get trapped at the pore openings. Furthermore, since KCTDs are expressed stronger than Cav2.3, two separate gels were necessary to avoid overexposure of KCTD bands.

5. If there is co-ip between CaV2.3 and KCTD8/12b, should there be actual co-localization observed in the EM images? Can this be measured?.

The reviewer raises a very important consideration. In freeze-fracture replica labeling, proteins that are closely associated with the membrane, such as transmembrane proteins or membrane-anchored proteins, get trapped in a carbon layer and remain there even after all remaining tissue is digested by SDS. Therefore, KCTDs, which are cytosolic proteins, are undetectable in replica unless they are covalently bound by chemical fixation to another membrane–integrated protein and getting trapped together in the carbon. As we readily detect them at high densities in the active zone, it very likely that the vast majority of detected KCTDs in replica are actually bound to either GBRs or Cav2.3.

Another important aspect is that it is difficult to label multiple proteins that are bound to each other in a complex because a) the protein complex might mask epitopes and b) primary/secondary antibodies with gold conjugates might hamper the binding of additional antibodies to another protein in the same complex in a process called steric hindrance.

Nevertheless, we made use of this steric hindrance effect to confirm the co-localization of KCTD8 and Cav2.3 in the active zone. We found that pre-incubation with anti-KCTD8 antibodies in KCTD12b KO mice led to a reduction in Cav2.3 labeling densities in the active zone of only ventral but not lateral MHb terminals. Although this does not prove that the two proteins are directly bound to each other, it suggests that they are in close proximity (<16 nm, two times IgG size of 8 nm) to each other. The following part was added to the Results section on page 16, starting from line 340:

“In order to examine further the spatial proximity of Cav2.3 to KCTD8, we tested whether steric hindrance of antibodies bound to KCTD8 might interfere with labeling of Cav2.3. We hypothesized that the compensatory increase in KCTD8 in the AZ might lead to increased binding of KCTD8 to Cav2.3 and, therefore, pre-incubation with anti-KCTD8 antibody might reduce Cav2.3 labeling in the AZ. In rostral IPN synapses, we found that pre-incubation significantly reduced Cav2.3 labeling densities in the AZ but not the peri- or extrasynaptic regions (Figure 7D; main effect of synaptic distribution: F_4, 880_ = 31.42, P < 0.0001; control AZ vs. pre-incubation AZ: P = 0.0006; two-way ANOVA with Bonferroni post hoc test). In the lateral IPN, pre-incubation significantly reduced Cav2.3 labeling density only in the peri-synaptic region (Figure 7E; main effect of synaptic distribution: F_4, 890_ = 39.01; control <50 nm vs. pre-incubation <50 nm: P = 0.0320).”

6. In Figure 8c, how selective is the augmentation with KCTD8 and CaV2.3, ie does it also occur for any other Ca channels? It is possible that a low level of other presynaptic Ca channels might be also augmented by KCTD8 in these terminals, increasing Ca^2+^ entry.

This is an interesting question and we do not have a definite answer to it. However, we did not observe any EPSC responses in rostral IPN neurons in Cav2.3 KO mice, suggesting that Cav2.3 is the sole Ca^2+^ channel mediating release at this pathway. Although the increase in release probability at ventral MHb terminals is likely caused by a direct effect of KCTD8 on Cav2.3, it is unknown whether other Ca^2+^ channel subtypes involved in transmitter release from dorsal MHb terminals are equally affected by KCTD8. We added this to the discussion on page 20, starting from line 435.

7. If this large augmentation of CaV2.3 currents by KCTD8 plays a physiological role should the epsc's in the KCTD8 KO mice be reduced? I cannot see this basic result, although I note the release probability is indeed reduced.

This is an important question. It is generally difficult to quantify absolute amplitudes of electrically evoked EPSCs as they depend largely on the stimulation intensity. Therefore, we used the variance mean analysis to show the reduced release probability in KCTD8 KO mice. An important indication that KCTD8 functionally increases synaptic strength was the observation that viral overexpression of KCTD8 in WT ventral MHb neurons significantly lowered PPR values compared with EGFP controls. This result suggests that KCTD8 and KCTD12b may be in a state of dynamic balance. Creating an imbalance appears to result in alterations in synaptic strength. The new data showing the effect of viral overexpression of KCTD8 on PPR was added to the Results section on page 14, starting from line 288 and discussed on page 20, starting from line 427.

8. Following on from point 6, in Discussion line 329, has the 1997 study by Ludwig et al. showing no α1B, A, C mRNA in the habenula been replicated anywhere? The technique used in this early study is not very sensitive, are there other more recent studies using single cell RNAseq for example?

Unfortunately, we did not find a more recent study investigating the expression of different Cav2 α subunits in MHb neurons. However, according to the Allen Brain Atlas, Cav2.1 and Cav2.2 are not expressed in MHb neurons. We included this in the supplement as Figure 8—figure supplement 1 and added this information to the discussion on page 18, starting from line 382.

9. Discussion line 330 "Although this exclusivity has been described anatomically (Parajuli et al., 2012)" is not correct as it gives the impression other Ca channels were examined in this study, whereas this is not the case.

We changed this sentence to: “Although the predominant presynaptic localization of Cav2.3 immunoreactivity was previously reported (Parajuli et al. 2012), the functional implications of this localization were not determined.” page 18, starting from line 385.

10. Discussion line 331: SNX482 is here described as specific, which is incorrect.

We removed the word “specific”.

Reviewer #3:The manuscript attempts to elucidate the mechanisms underlying the potentiation of synaptic transmission between the ventral medial habenula (MH) and the rostral interpeduncular nucleus mediated by presynaptic GABA-B receptors. The topic is interesting especially because it is very unusual that GABA-B receptors gate an enhancement of neurotransmission, since usually when they are activated, they depress the release of neurotransmitter.The manuscript contains some high quality data, for example, freeze-fracture replica immunolabelling of presynaptic proteins in habenular terminals. However, I also found some technical problems, (e.g., see comment #1 below). Overall, the manuscript is potentially interesting and covers an original topic. However, the data currently submitted fail to answer the two major questions, namely the underlying mechanisms and the physiological relevance of GABA-B receptor-mediated modulation. A lot of efforts are devoted in characterizing key protein expression in various K^+^ channel tetramerization domain-containing (KCTD) proteins. However, I doubt that this strategy paid off and really shed light on the mechanisms underlying GABA-B receptor mediated modulation of the synapses studied.

We agree with the reviewer that we failed to elucidate the mechanisms of GABA-B receptor-mediated modulation in our previous manuscript. Also, according to suggestion by the reviewer #2 (please see above), we have re-structured our manuscript mostly focusing on the role of KCTDs on the Cav2.3-mediated basal neurotransmission in the MHb-IPN pathway. We will deal with the mechanism of GABA_B_ receptor-mediated potentiation in a separate paper.

1. Data on the action of SNX-482 on evoked EPSCs need refinement. The presynaptic fibers of the fasciculus retroflexus were electrically stimulated and recordings were made in the rostral IPN. In contrast, stimulation of dorsal fibers were obtained via optogenetic stimulation and responses were recorded in the lateral IPN. The effect of SNX-482 was stronger at the former versus the latter synapses. However, this result is at odds with similar Cav2.3 densities in the active zones of dorsal and ventral MHb terminals. The Authors discussed this issue, and I acknowledged this mis-match in their data. Nevertheless, the problem remains: a quantitative comparison between drug modulation of synaptic responses evoked by electrical versus optogenetic stimulation is a poor choice. Amongst various problems, several recent papers report the non-physiological high release probability often caused by optogenetic stimulation of presynaptic fibers that contrasts with the electrical stimulation method. This discrepancy would affect the results obtained with SNX-482. This set of experiments should be repeated by using more uniform experimental conditions in the two synaptic pathways.

We agree that using two separate ways of stimulation for the two pathways was not ideal. We initially used optogenetics to study lateral IPN inputs because it was challenging to keep the fiber tract intact all the way to the lateral IPN. This is due to the fact that although MHb axons project linearly from the MHb to the IPN, they enter the IPN from the anterior side. Thus, to solve this problem, we have newly established thick-slice preparations (1 mm) cut at a 54° angle, which keep the most anterior portion of the IPN intact within the slice and allow reliable stimulation of the fasciculus retroflexus (FR). In this preparation we stimulated FR just below the MHb, 2 – 3 mm distal to the IPN, and could obtain EPSC responses in both rostral and lateral IPN neurons (see Figure 1—figure supplement 1B – D). We used a 10 Hz stimulus train rather than a single stimulus in order to not miss potentially weaker responses due to the absence of Cav2.3. The following was added to the Results section on page 6 starting from line 112:

“To corroborate the involvement of Cav2.3 in release from MHb terminals, we next performed electrical stimulation of the MHb-derived fiber tract, the fasciculus retroflexus (FR), in Cav2.3 KO mice and recorded EPSCs in rostral and lateral IPN neurons. To obtain maximally intact FR, slices were cut at 1 mm thickness which allowed for the stimulation of the fiber tract at a distance of 2 – 3 mm from the IPN (Figure 1H, I). […] Overall, these results suggest that ventral MHb terminals rely exclusively on Cav2.3 for release, whereas dorsal MHb terminals rely on Cav2.3 and additional Ca^2+^ channels.”

Furthermore, a serious comparison of the efficacy of SNX-482 should rely on testing at least three different drug concentrations for each synapse. Furthermore, normalization assuring that approximatively an equal number of presynaptic fibers are stimulated at each synaptic connection should be employed. What ever methods of presynaptic stimulation will be used, optical or electrical, test of selective stimulation of fibers should be also performed.

As described in our response to the reviewer’s initial concern (1.), we tested the effect of SNX-482 using fiber tract electrical stimulation in WT mice and directly compared EPSC responses in rostral and lateral IPN neurons in the same slice of Cav2.3 KO mice using electrical fiber tract stimulation. Stimulation selectivity was confirmed by application of baclofen in WT slices (Figure 1—figure supplement 1C, D), which potentiated EPSCs when recording from rostral IPN neurons and inhibited EPSC when recording from lateral IPN neurons. We also compared the time courses of SNX-482-mediated EPSC reductions in lateral IPN for electrical and optogenetic stimulation in Figure 1L, confirming no significant difference between types of stimulation and no effect of Conotoxin application on EPSC amplitudes. This point was also added to the Results section on page 7, starting from line 136.

2. Baclofen still presynaptically potentiated EPSCs in KCTD KO mice. However, loss of KCTD12b impaired GABABR action probably via increasing release probability and occlusion. In contrast, lack of KCTD8 reduces release probability. But what about the effects of GABABR modulation? Is baclofen more powerful at MH synapses in KCTD8 KO?

We did not find a significant difference in baclofen-mediated potentiation between WT and KCTD8 KO mice, suggesting that the relatively minor reduction in release probability in KCTD8 KO mice did not affect GBR-mediated enhancement of release. Since the reviewers felt that our manuscript focused too much on GBR-mediated enhancement instead of the effect of KCTDs on basal release, we removed discussion and analysis of the potential connection between basal release and baclofen-mediated enhancement. We now only mention that GBR-mediated potentiation was still intact in all KCTD KO mice and moved the corresponding result to the supplement (Figure 6—figure supplement 1).

3. The manuscript lacks to experimentally demonstrate that GABA-B receptor potentiation of ventral MH and the rostral interpeduncular nucleus synapses is physiologically relevant. This should be addressed by using physiologically salient repetitive stimulation in control and in the presence of GABA-B R antagonists. It would be also interesting to test the GABA-B R ligands on evoked EPSPs followed by IPSPs to see the comprehensive pharmacological effects on both excitation and inhibition. At present none of these important points are addressed and this reduces the impact of the data currently submitted.

We were unfortunately unable to unravel the exact mechanism underlying the GBR-mediated enhancement of release in our original manuscript but instead found a surprising involvement of KCTDs with Cav2.3 that modulate basal release. As the reviewers suggested, we therefore shifted our focus away from potential GBR mechanisms, and now mostly focus on the role of KCTDs in basal release.

[Editors’ note: what follows is the authors’ response to the second round of review.]

The revised manuscript has been significantly improved by refocusing the study on the role for KCTDs on neurotransmitter release. Several additional experiments have been performed, including the comparisons of Cav2.3-dependence of synaptic transmission between the two distinct medial habenula to IPN inputs by electrical stimulation, which has been made possible by the establishment of thick-slice preparation. New data lend further support to the conclusion that KCTD8 and KCTD12b directly interact with Cav2.3 and regulate release from ventral MHb terminals independently of GABA_B_ receptors, where effectively KCTD12b serves to restrain release while the recruitment of KCTD8 to the active zone potentiates release by promoting Cav2.3 currents. Prior to accepting the manuscript for publication, a number of points require attention.Figure 6F-G: the EPSC waveforms of some rescued currents appear to be quite different than one would expect. Specifically, the LV-hSyn-KCTD8 current is quite prolonged, and may have two components to the rising phase.

We agree that the LV-hSyn-KCTD8 trace appears slightly broader than the control trace. Upon re-examination of the other recordings in this group, we did not consistently observe broader responses in KCTD8 overexpression experiments and therefore, replaced the corresponding example trace with one showing a more representative decay time.

Additionally, the LV-hSyn-KCTD12b current appears to have a delayed onset. Are these examples representative of the population? If so, are these effects a result of over-expression, or more an indicator of the distinct roles of these two KCTD proteins?

Concerning the slightly delayed onset in the KCTD12b rescue trace: This delay might be influenced by the position of the stimulating electrode relative to the recorded neuron. Inside the IPN, habenula-derived axons can be unmyelinated and therefore responses may occur slightly faster or slower depending on this distance. Although we try to keep recording conditions similar for all experiments, random biological variation cannot be completely controlled. Upon re-examination of the other recordings, we did not consistently observe delayed responses in LV-hSyn-KCTD12b injected mice, suggesting a KCTD-independent effect, and we replaced the example trace with one showing a more representative response onset.

In the introduction (Line 47 or Line 53-54), it would be clearer to readers who may not necessarily be familiar with KCTDs if the authors could briefly state the nature of the molecular difference especially between KCTD12 and KCTD12b.

In accordance with this suggestion, we rewrote the second paragraph of the introduction:

“Four KCTD subunits, KCTD8, 12, 12b and 16, serve as auxiliary GBR subunits (Schwenk et al., 2010). KCTD subunits bind as hetero- and homo-pentamers to GBRs and modulate their signaling kinetics (Fritzius et al. 2017; Zheng et al., 2019). Structurally, KCTD subunits are composed of an N-terminal T1 domain and an H1 domain. […] However, the functional consequences of these interactions and whether other voltage-gated Ca^2+^ channels interact with KCTDs remain unknown.”

Line 302-304. "The decrease in release probability by KCTD12b in the presence but not in the absence of KTCD8…" is confusing. Do the authors mean that the reason why the decreased PPR observed in KCTD12bKO could be recovered to control levels when KCTD8 is additionally absent (as observed in double KO mice), could be because the loss of KCTD12b in itself alters the localization of KCTD8 or Cav2.3? Perhaps this sentence should be rephrased.

We apologize for the confusing wording, we rephrased the sentence as follows: “The decreased PPR in the absence of KCTD12b recovered to control levels when KCTD8 was additionally absent suggesting alterations in either Cav2.3 or KCTD8 localization in the active zone of MHb terminals in KCTD12b KO mice.”